# Stochastic Schrödinger Diffusion Models for Pure-State Ensemble Generation

## Abstract

In quantum machine learning (QML), classical data are often encoded as quantum pure states and processed directly as quantum representations, motivating *representation-level generative modeling* that samples new quantum states from an underlying pure-state ensemble rather than re-preparing them from perturbed classical inputs. However, extending *score-based* diffusion models with well-defined reverse-time samplers to quantum pure-state ensembles remains challenging, due to the non-Euclidean geometry of the complex projective space $\mathbb{CP}^{d-1}$ and the intractability of transition densities. We propose *Stochastic Schrödinger Diffusion Models* (SSDMs), an intrinsic score-based generative framework on $\mathbb{CP}^{d-1}$ endowed with the Fubini–Study (FS) metric. SSDMs formulate a forward Riemannian diffusion with a stochastic Schrödinger equation (SSE) realization, and derive reverse-time dynamics driven by the Riemannian score $\nabla_{\mathrm{FS}} \log p_t$. To enable training without analytic transition densities, we introduce a local-time objective based on a local Euclidean Ornstein–Uhlenbeck approximation in FS normal coordinates, yielding an analytic teacher score mapped back to the manifold. Experiments show that SSDMs faithfully capture target pure-state ensemble statistics, including observable moments, overlap-kernel MMD, and entanglement measures, and that SSDM-generated quantum representations improve downstream QML generalization via representation-level data augmentation.

## 1. Introduction

Diffusion and score-based generative models have recently emerged as a dominant paradigm for learning complex data distributions (Yang et al., 2022; Chen et al., 2025), achieving state-of-the-art performance in domains ranging from images (Song et al., 2021; Li et al., 2025) and audio (Chen et al., 2021; Kong et al., 2021) to molecules (Hoogeboom et al., 2022; Xu et al., 2022) and protein structures (Watson et al., 2023; Trippe et al., 2023; Wu et al., 2024). These methods rely on a simple idea: define a tractable stochastic *forward* process that gradually destroys structure, then learn to *reverse* this process by estimating the score function, i.e., the gradient of the log-density along the diffusion trajectory (Song et al., 2021; Ho et al., 2020). The resulting reverse-time dynamics (Anderson, 1982; Haussmann & Pardoux, 1986; Song et al., 2021) enables sampling from an otherwise intractable data distribution.

Motivated by the increasing role of quantum representations in quantum machine learning (QML) (Biamonte et al., 2017; Preskill, 2018), a natural question arises: *can score-based diffusion be extended into a practical generative modeling framework for quantum* representations?

In many QML settings, classical inputs are encoded as quantum *pure states*—via amplitude/phase embeddings, variationally prepared feature maps, or as intermediate states of quantum algorithms—and downstream models operate directly on these state representations (Schuld & Killoran, 2019; Havlíček et al., 2019). When available quantum data are scarce (a common scenario in near-term experiments) (Preskill, 2018), one would like to augment or simulate such quantum representations by sampling additional states from the underlying ensemble.

However, naïvely perturbing the classical input space and re-encoding can easily produces pathological quantum states: small changes in $x$ can lead to nearly orthogonal feature states, collapsed entanglement structure, or samples that concentrate on measure-zero regions of the state space, ultimately harming training stability and generalization (Havlíček et al., 2019; Huang et al., 2021). This motivates generative modeling *directly in the space of pure states*: learning an implicit distribution $p_0(|\psi\rangle)$ from observed quantum representations and *sampling new pure states from the resulting ensemble distribution* could enable representation-level data augmentation, distributional modeling, and efficient simulation of quantum feature ensembles.

[1]Anonymous Institution, Anonymous City, Anonymous Region, Anonymous Country. Correspondence to: Anonymous Author <anon.email@domain.com>.

Preliminary work. Under review by the International Conference on Machine Learning (ICML). Do not distribute.

Yet extending diffusion models to this setting is nontrivial. Quantum pure states do not form an unconstrained Euclidean space: they live on the complex projective manifold $\mathbb{CP}^{d-1}$ modulo global phase and are endowed with the Fubini–Study metric (Mielnik, 1968). Consequently, the design of forward diffusions, the definition of the score, and the construction of reverse-time samplers require careful geometric and physical considerations.

Recent work has begun exploring "quantum diffusion" from multiple perspectives. Some approaches define forward diffusion as a noisy quantum channel and train a parameterized inverse process for state recovery or state preparation (Chen et al., 2024; Parigi et al., 2025; Zhang et al., 2023; Zhu et al., 2025). Others relate measurement-induced stochastic trajectories to diffusion-like dynamics and characterize corresponding reverse processes from a physical recovery viewpoint (Dalibard et al., 1992; Gisin & Percival, 1992; Kiefer, 2010). While these studies provide valuable connections between stochastic quantum dynamics and diffusion, a *practical score-based generative modeling framework* with *well-defined reverse-time sampling dynamics* for quantum pure-state ensembles remains underdeveloped. In particular, existing formulations often emphasize recovery of a specific state or trajectory, or focus on density-matrix evolution, whereas QML applications frequently require *sampling new pure-state instances* from an implicit ensemble distribution.

In this work, we propose *Stochastic Schrödinger Diffusion Models* (SSDMs), a score-based generative framework for distributions over quantum pure states. Our central idea is to construct an intrinsic diffusion directly on the pure-state manifold $\mathbb{CP}^{d-1}$ endowed with the Fubini–Study (FS) metric, building on recent advances in score-based diffusion beyond Euclidean spaces (Bortoli et al., 2022; Huang et al., 2022). Concretely, we design a *Riemannian Ornstein–Uhlenbeck (OU)* forward diffusion that is stable and tractable, and we provide a stochastic Schrödinger (Bouten et al., 2004) realization whose projective dynamics induces the same manifold process (Gisin & Percival, 1992; Kiefer, 2010). In FS normal coordinates, this diffusion reduces to a first-order OU/VP (variance-preserving, (Song et al., 2021)) process with curvature corrections, establishing a rigorous bridge to Euclidean score-based diffusion (Hsu, 2002).

Building on the theory of time reversal for (manifold) diffusions, we derive the corresponding reverse-time dynamics on $\mathbb{CP}^{d-1}$ and show that it involves a *Riemannian score* term defined with respect to the FS geometry (Anderson, 1982; Haussmann & Pardoux, 1986; Bortoli et al., 2022; Huang et al., 2022). This yields a principled generalization of Euclidean score-based generative modeling to quantum pure-state ensembles.

A key technical challenge is that the transition density of the forward diffusion on $\mathbb{CP}^{d-1}$ is generally not available in closed form, which prevents direct evaluation of the score (Huang et al., 2022; Lou et al., 2023b). To address this, we develop a *local-time learning objective* that exploits a fundamental geometric fact: in Fubini–Study normal coordinates, the intrinsic manifold diffusion admits a *local Euclidean OU/VP approximation* to first order, with curvature effects entering only at higher order (Hsu, 2002). This yields a closed-form Gaussian surrogate for short-time transitions in the tangent space, from which we obtain an analytic teacher score. This local construction is essential because global transition densities and scores on $\mathbb{CP}^{d-1}$ are intractable, whereas short-time behavior is governed by universal geometric structure that can be exploited for stable learning. We then map the teacher score back to the manifold via the differential of the logarithm map and distill these local conditional score signals into a global score model (Bortoli et al., 2022; Huang et al., 2022). In practice, we train a score network (classical or quantum) to approximate the Riemannian score field, and we sample new quantum pure states by integrating the learned reverse-time dynamics from a simple prior distribution. For the prior, we consider hardware-friendly choices such as Clifford $t$-design ensembles, which provide efficient approximations to Haar-random sampling (Dankert et al., 2009; Zhu, 2017).

We evaluate SSDMs on generative modeling tasks over quantum pure-state ensembles, including ensembles induced by quantum feature embeddings of classical data and variational state families (Schuld & Killoran, 2019; Havlíček et al., 2019). We assess distribution matching via observable statistics, kernel MMD (Gretton et al., 2012), and entanglement-related measures (Mielnik, 1968), and we demonstrate that generated quantum encodings can serve as effective data augmentation to improve downstream QML generalization.

Our main contributions are:

- We introduce *Stochastic Schrödinger Diffusion Models* (SSDMs), a *score-based* generative modeling framework for *sampling ensemble distributions* over quantum pure states on $\mathbb{CP}^{d-1}$.

- We design an intrinsic *Riemannian OU* forward diffusion on $\mathbb{CP}^{d-1}$ and provide an *SSE realization*; we derive the associated reverse-time dynamics involving a *Riemannian score* under the FS metric.

- We propose a practical *local-time* learning objective that leverages the *local Euclidean OU limit* in FS normal coordinates to obtain an analytic Gaussian teacher score for short-time transitions, which is mapped back to the manifold via the logarithm map and enables Riemannian denoising score matching without requiring analytic transition densities.

- We demonstrate empirically that SSDMs generate high-quality samples matching target ensemble statistics and provide measurable benefits for QML applications such as quantum data augmentation.

## 2. Background

### 2.1. Score-Based Diffusion Models in Euclidean Space

Score-based diffusion models construct a family of distributions $\{p_t\}_{t \in [0,T]}$ by running a simple forward noising SDE that transports data to a tractable prior (Song et al., 2021):

$$dx_t = f(x_t, t) \, dt + g(t) \, dw_t, \tag{1}$$

where $w_t$ is standard Brownian motion. A key result is that the reverse-time dynamics is also an SDE whose drift involves the score $\nabla_x \log p_t(x)$ (Song et al., 2021):

$$dx_t = \big( f(x_t, t) - g(t)^2 \nabla_x \log p_t(x_t) \big) \, dt + g(t) \, d\bar{w}_t. \tag{2}$$

In practice, a neural network $s_\theta(x, t) \approx \nabla_x \log p_t(x)$ is learned via denoising score matching, and sampling is performed by integrating (2) from the prior $p_T$ back to $t = 0$.

A common choice of forward process is the OU/VP diffusion, which yields a stable Gaussian prior and supports efficient discretizations. Our goal is to develop an analogous framework for quantum pure states, where the state space is the curved manifold $\mathbb{CP}^{d-1}$ rather than $\mathbb{R}^d$.

### 2.2. Quantum Pure States and the Fubini–Study Geometry

Let $\mathcal{H} \cong \mathbb{C}^d$ be a $d$-dimensional Hilbert space. A quantum pure state is represented by a unit vector $|\psi\rangle \in \mathcal{H}$ with $\langle \psi, \psi \rangle = 1$. Physical states are invariant under global phase, i.e., $|\psi\rangle \sim e^{i\phi} |\psi\rangle$. The space of pure states is therefore the complex projective space

$$\mathcal{M} := \mathbb{CP}^{d-1}. \tag{3}$$

The natural Riemannian metric on $\mathcal{M}$ is the Fubini–Study (FS) metric, which induces the geodesic distance

$$d_{\text{FS}}(\psi, \phi) = \arccos \big| \langle \psi, \phi \rangle \big|. \tag{4}$$

This geometry plays a central role in our construction: diffusion takes place on the manifold $\mathcal{M}$ and the relevant score is a *Riemannian score* defined with respect to the FS metric.

### 2.3. Stochastic Schrödinger Equations and Quantum Trajectories

Open quantum systems are often modeled by Lindblad master equations (Manzano, 2020) for density matrices $\rho_t$,

$$\frac{d}{dt}\rho_t = -i[H, \rho_t] + \sum_k \gamma_k \Big( L_k \rho_t L_k^\dagger - \tfrac{1}{2} \{ L_k^\dagger L_k, \rho_t \} \Big), \tag{5}$$

where $H$ is the system Hamiltonian and $\{L_k\}$ are Lindblad operators. A key concept is *unraveling*: the mixed-state $\rho_t$ evolution in (5) can be represented as an ensemble average over stochastic pure-state trajectories $|\psi_t\rangle$ governed by a Stochastic Schrödinger Equations (SSE) (Bouten et al., 2004). A generic Itô-form SSE can be written as

$$d \, |\psi_t\rangle = A(|\psi_t\rangle, t) \, dt + \sum_k B_k(|\psi_t\rangle, t) \, dw_t^{(k)}, \tag{6}$$

where $\{w_t^{(k)}\}$ are independent Wiener processes and the drift/diffusion terms are chosen such that $\mathbb{E}\big[ |\psi_t\rangle \langle \psi_t| \big] = \rho_t$. In this work, we use SSEs not only as a physical motivation but as a *design principle* for defining a tractable forward diffusion over pure states.

### 2.4. Diffusions on Manifolds and Riemannian Scores

Let $(\mathcal{M}, g)$ be a Riemannian manifold and let $p_t$ denote the time-marginal density with respect to the Riemannian volume measure. The natural analog of the Euclidean score is the *Riemannian score*

$$\nabla_g \log p_t(\psi) \in T_\psi \mathcal{M}, \tag{7}$$

where $\nabla_g$ is the Riemannian gradient. For manifold diffusions, time reversal admits an intrinsic formulation in which the reverse drift contains the Riemannian score, mirroring the Euclidean case (Huang et al., 2022; De Bortoli et al., 2022).

In general, transition densities on $\mathcal{M}$ are intractable (Lou et al., 2023a). We therefore leverage a local approximation: in normal coordinates, an isotropic manifold diffusion is Euclidean to first order, so over a small time step it behaves like an OU/VP update in the tangent space, with curvature effects entering at higher order. This yields a Gaussian surrogate for the short-time conditional density and an analytic teacher score, which we map back to the manifold via the differential of the logarithm map and use to train a global score model for reverse-time sampling.

## 3. Stochastic Schrödinger Diffusion Models

We propose *Stochastic Schrödinger Diffusion Models* (SSDMs), a score-based generative framework for learning and sampling *distributions over quantum pure states*. A pure state $|\psi\rangle \in \mathcal{H} \cong \mathbb{C}^d$ is physically represented by its equivalence class $[\psi] \in \mathcal{M} := \mathbb{CP}^{d-1}$, the complex projective space endowed with the Fubini–Study (FS) metric $g_{\text{FS}}$. SSDMs perform diffusion modeling *intrinsically* on $(\mathcal{M}, g_{\text{FS}})$: we define a tractable *forward* noising diffusion that maps an unknown data ensemble $p_0$ to a simple base distribution $p_T$, and learn a *reverse-time* diffusion whose drift is driven by the *Riemannian score* $\nabla_{\text{FS}} \log p_t$. The main technical obstacle is that transition densities on $\mathbb{CP}^{d-1}$ are generally

unavailable in closed form, so we introduce a *local-time training objective* that uses a short-time Gaussian approximation in FS normal coordinates to provide an analytic teacher score.

### 3.1. Forward Diffusion on $\mathbb{CP}^{d-1}$ with an SSE Realization

**Intrinsic manifold diffusion.** We construct a continuous-time forward diffusion $\{\psi_t\}_{t \in [0,T]}$ on $\mathcal{M}$ whose marginals gradually transform $p_0$ into a tractable base distribution $p_T$. We model the forward process as a time-inhomogeneous diffusion on $(\mathcal{M}, g_{\mathrm{FS}})$:

$$d\psi_t = a(\psi_t, t)\, dt \; + \; \sigma(t)\, dW_t^{(\mathcal{M})}, \tag{8}$$

where $W_t^{(\mathcal{M})}$ denotes Brownian motion on $\mathcal{M}$ under the FS metric and $a(\psi, t)$ is a (possibly zero) drift. Throughout, we choose $a$ and $\sigma$ so that the noising is *approximately isotropic* under $g_{\mathrm{FS}}$ and progressively destroys the structure of $p_0$. When $a \equiv 0$, Eq. (8) reduces to FS-Brownian motion (heat flow), whose long-time limit is the unitarily-invariant base measure on $\mathcal{M}$.

**OU-style discretization and tangent projection.** For implementation, it is convenient to express the diffusion using tangent vector fields and a projection that enforces the projective-state constraint. We consider the Stratonovich-form update

$$d\psi_t = \mathcal{P}_{\psi_t}\Big(b(\psi_t, t)\, dt + \sigma(t) \sum_{k=1}^{K} V_k(\psi_t) \circ dw_t^{(k)}\Big), \tag{9}$$

where $\{w_t^{(k)}\}_{k=1}^{K}$ are independent Wiener processes, $V_k(\psi)$ are tangent vector fields, and $\mathcal{P}_\psi$ projects ambient increments onto $T_\psi \mathcal{M}$, ensuring invariance to global phase. With an appropriate choice of $\{V_k\}$ (e.g., an orthonormal frame under $g_{\mathrm{FS}}$) and drift $b$, Eq. (9) induces the intrinsic diffusion (8) up to higher-order curvature effects. See Appendix A for a formal statement.

**SSE realization (stochastic unitary evolution).** To connect the manifold diffusion with physics-motivated dynamics, we provide an SSE realization on the Hilbert sphere whose induced process on $\mathbb{CP}^{d-1}$ is approximately isotropic. Let $\{G_k\}_{k=1}^{d^2-1}$ be an orthonormal basis of $\mathfrak{su}(d)$ (e.g., generalized Gell–Mann matrices (Loubenets, 2020)). We consider the Stratonovich SDE (Kurtz et al., 1995)

$$d\,|\psi_t\rangle = -\, iH(t)\, |\psi_t\rangle\, dt \; - \; \frac{1}{2}\eta(t) \sum_{k} G_k^2\, |\psi_t\rangle\, dt$$
$$- \; i\sqrt{\eta(t)} \sum_{k} G_k\, |\psi_t\rangle \circ dW_t^{(k)}, \tag{10}$$

whose stochastic unitary term generates diffusion along Lie-algebra directions. After quotienting out the global phase, the induced dynamics on $\mathbb{CP}^{d-1}$ matches the generator of (8) up to curvature-dependent terms; details are given in Appendix B.

**Prior distribution.** We initialize reverse-time sampling from a tractable base distribution $p_T$ on $\mathcal{M}$, with the isotropic FS/Haar measure as the ideal choice. In practice, we approximate it using ensembles such as Clifford $t$-designs or other tractable alternatives.

### 3.2. Reverse-Time Dynamics and the Riemannian Score

Given the forward diffusion on the pure-state manifold $(\mathcal{M}, g_{\mathrm{FS}})$ defined in Eq. (8), let $p_t(\psi)$ denote its time-marginal density with respect to the Riemannian volume measure induced by the FS metric. For the Riemannian OU forward process,

$$d\psi_t = b(\psi_t, t)\, dt + \sigma(t)\, dW_t^{(\mathcal{M})},$$
$$b(\psi, t) := -\, \lambda(t) \operatorname{Log}_\psi(\psi_\star), \tag{11}$$

the associated (time-inhomogeneous) generator takes the form

$$\mathcal{L}_t f(\psi) = \langle b(\psi, t), \nabla_{\mathrm{FS}} f(\psi)\rangle_{\mathrm{FS}} + \frac{\sigma(t)^2}{2}\Delta_{\mathrm{FS}} f(\psi), \tag{12}$$

where $\Delta_{\mathrm{FS}}$ is the Laplace–Beltrami operator on $\mathbb{CP}^{d-1}$. See Proposition C.1 in Appendix C.

**Reverse-time diffusion on $\mathcal{M}$.** A standard result on time reversal of diffusions on Riemannian manifolds implies that the reverse-time process is again a diffusion on $\mathcal{M}$ with the same diffusion coefficient but a modified drift. In intrinsic Stratonovich form, the reverse-time dynamics can be written as

$$d\psi_t = \tilde{b}(\psi_t, t)\, dt + \sigma(t)\, d\bar{W}_t^{(\mathcal{M})},$$
$$\tilde{b}(\psi, t) = b(\psi, t) - \sigma(t)^2\, \nabla_{\mathrm{FS}} \log p_t(\psi). \tag{13}$$

A precise statement and proof are provided in Proposition C.2 in Appendix C; see Fig. 1 for an illustration of the forward and reverse diffusion processes.

**Connection to our SSE realization.** Since our forward diffusion admits an SSE realization in Stratonovich form (Eq. (10)), Eq. (13) provides a principled reverse-time sampler for Schrödinger-type diffusions on $\mathbb{CP}^{d-1}$. In practice, we approximate the score $s^\star(\psi, t)$ with a parameterized model $s_\theta(\psi, t)$ and integrate the learned reverse dynamics from $\psi_T \sim p_T$ to obtain samples at $t = 0$.

**Remark (coordinate corrections).** Eq. (13) is stated intrinsically in Stratonovich form. When rewritten in local

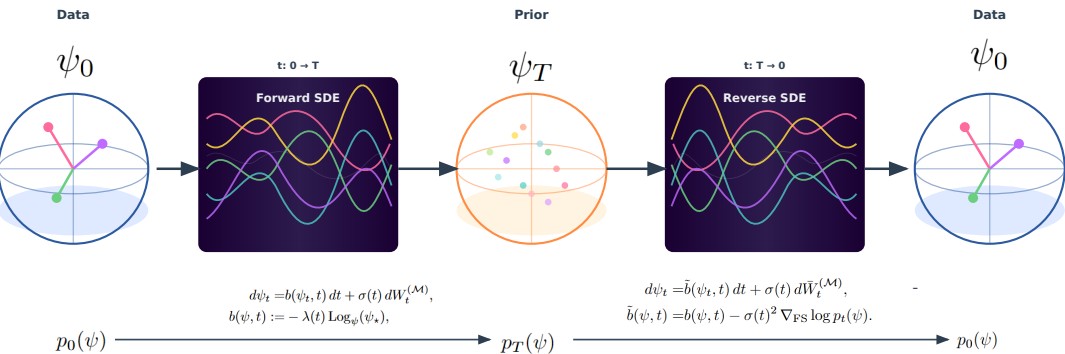

**Figure 1.** Illustration of the forward diffusion on the quantum pure-state manifold and its reverse-time generative process driven by a learned Riemannian score.

coordinates or converted to Itô form, additional geometry-dependent correction terms appear (e.g., Levi-Civita connection and divergence terms associated with the Riemannian volume measure). In our implementation, we perform updates in local orthonormal frames on $T_\psi \mathcal{M}$ and map tangent increments back to $\mathcal{M}$ via $\mathrm{Exp}$ (or a retraction), which avoids the need to explicitly write these coordinate corrections; we provide the coordinate-form expressions in Appendix D.

### 3.3. Local-Time Teacher Scores via FS Normal Coordinates

A central challenge is that the marginal density $p_t(\psi)$ is not available in closed form, which prevents direct evaluation of the Riemannian score $\nabla_{\mathrm{FS}} \log p_t(\psi)$. We therefore construct a *local-time teacher score* based on the fact that the forward diffusion admits a *local Euclidean OU limit* in Fubini–Study (FS) normal coordinates.

**Local analytic teacher score via FS normal coordinates.** Fix a short step size $\delta t > 0$. Given a local-time pair $(\phi, \psi) := (\psi_{t-\delta t}, \psi_t)$ from the forward process (11), define the FS normal coordinates centered at $\phi$ by

$$z := \log_\phi(\psi) \in T_\phi \mathcal{M}. \quad (14)$$

Here, $\log_\phi(\psi)$ denotes the Riemannian logarithm map that expresses $\psi$ as a tangent vector at the base point $\phi$, i.e., the initial velocity of the unique geodesic starting from $\phi$ and reaching $\psi$. As discussed in Sec. 3.2, in these coordinates the forward manifold diffusion is well-approximated, for sufficiently small $\delta t$, by an Euclidean OU/VP step,

$$z \approx \alpha(t, \delta t)\, z_{t-\delta t} + \beta(t, \delta t)\, \xi, \qquad \xi \sim \mathcal{N}(0, I), \quad (15)$$

where $\alpha, \beta$ are determined by the diffusion schedule and curvature effects enter only at higher order in $\|z\|$ (equivalently,

higher order in $\delta t$ in the small-step regime). For the local-time construction we take $z_{t-\delta t} = 0$ (since $\log_\phi(\phi) = 0$), so the induced *local conditional density* $q(z \mid 0)$ is Gaussian with covariance $\Sigma(t, \delta t) = \beta(t, \delta t)^2 I$ in the isotropic OU/VP case. This yields a closed-form teacher score in normal coordinates:

$$s_z^{(\mathrm{teach})}(z, t, \delta t) := \nabla_z \log q(z \mid 0) = -\Sigma(t, \delta t)^{-1} z. \quad (16)$$

We then map this teacher score back to the manifold using the adjoint of the differential of the logarithm map:

$$s^{(\mathrm{teach})}(\psi, \phi, t, \delta t) := (d \log_\phi)^*_\psi\, s_z^{(\mathrm{teach})}(z, t, \delta t),$$
$$z = \log_\phi(\psi), \quad (17)$$

Here $(d \log_\phi)^*_\psi$ denotes the adjoint of the differential of the logarithm map with respect to the Riemannian metric, mapping cotangent vectors back to $T_\psi \mathcal{M}$. This adjoint naturally arises since the score is a gradient and therefore transforms via the adjoint of the Jacobian under coordinate changes. In FS normal coordinates, this construction approximates the intrinsic local conditional score $\nabla_{\mathrm{FS}} \log p(\psi \mid \phi)$ up to curvature and volume-element corrections of order $O(\|z\|^2)$; see Proposition E.1 in Appendix E.

### 3.4. Riemannian Denoising Score Matching Objective

We train a parameterized score model $s_\theta(\psi, t)$ to approximate the Riemannian score $s^\star(\psi, t) = \nabla_{\mathrm{FS}} \log p_t(\psi)$ on $\mathcal{M} = \mathbb{CP}^{d-1}$. Given data samples $\psi_0 \sim p_0$, we simulate the forward diffusion and sample a random time $t \sim \mathcal{U}(0, T)$. For a small local step $\delta t$, we form the local-time pair

$$(\phi, \psi) := (\psi_{t-\delta t}, \psi_t), \quad (18)$$

and compute the teacher score $s^{(\mathrm{teach})}(\psi, \phi, t, \delta t)$ via the analytic local OU approximation in FS normal coordinates

(Eqs. (16)–(17)).

We then minimize a Riemannian denoising score matching objective:

$$\mathcal{L}(\theta) = \mathbb{E}_{\psi_0 \sim p_0,\, t \sim \mathcal{U}(\delta t, T),\, (\phi, \psi) \sim p(\psi_{t-\delta t}, \psi_t | \psi_0)} \Big[ \lambda(t, \delta t)$$
$$\big\| \mathcal{P}_\psi \big( s_\theta(\psi, t) \big) - s^{(\text{teach})}(\psi, \phi, t, \delta t) \big\|_{\text{FS}}^2 \Big].$$
$$(19)$$

where $\| \cdot \|_{\text{FS}}$ denotes the norm induced by the FS metric on the tangent space and $\mathcal{P}_\psi$ projects a vector onto $T_\psi \mathcal{M}$. Unless stated otherwise, we use $\lambda(t, \delta t) = \beta(t, \delta t)^2$, which mirrors the variance-weighting commonly used in denoising score matching for VP diffusions.

This objective distills local short-time conditional score information into a global score estimator that can be used for reverse-time sampling over the full diffusion horizon. At the population level, minimizing the above objective recovers the marginal Riemannian score $s^\star(\psi, t) = \nabla_{\text{FS}} \log p_t(\psi)$; see Proposition F.1. Moreover, using the local-time teacher score is consistent in the small-step limit $\delta t \to 0$; see Proposition F.2 in Appendix F.

### 3.5. Sampling Algorithm

After training the score model $s_\theta(\psi, t) \approx \nabla_{\text{FS}} \log p_t(\psi)$, we generate samples by drawing $\psi_T \sim p_T$ from a tractable prior (e.g., a Clifford $t$-design ensemble) and integrating the learned reverse-time dynamics on $\mathcal{M} = \mathbb{CP}^{d-1}$.

**Reverse-time sampling SDE.** Using Eq. (13), we simulate the reverse diffusion

$$d\psi_t = \big( b(\psi_t, t) - \sigma(t)^2 s_\theta(\psi_t, t) \big) dt + \sigma(t)\, d\bar{W}_t^{(\mathcal{M})}, \quad (20)$$

where $b(\psi, t) = -\lambda(t) \text{Log}_\psi(\psi_\star)$ for the Riemannian OU forward process and $\bar{W}_t^{(\mathcal{M})}$ denotes reverse-time Brownian motion on $(\mathcal{M}, g_{\text{FS}})$.

**Manifold discretization.** Let $t_k$ be a discretization of $[0, T]$ with step $\Delta t = t_k - t_{k-1}$ (integrated backward from $T$ to 0), and define $\tau_k := \int_{t_{k-1}}^{t_k} \sigma(s)^2 ds \approx \sigma(t_k)^2 \Delta t$. We update the state using an Euler–Maruyama step in the tangent space followed by a retraction onto $\mathcal{M}$:

$$\xi_k \sim \mathcal{N}(0, I) \quad \text{(in a local orthonormal basis of } T_{\psi_{t_k}} \mathcal{M}),$$
$$v_k = \big( b(\psi_{t_k}, t_k) - \sigma(t_k)^2 s_\theta(\psi_{t_k}, t_k) \big) \Delta t + \sqrt{\tau_k}\, \xi_k,$$
$$\psi_{t_{k-1}} = \text{Exp}_{\psi_{t_k}}(v_k),$$
$$(21)$$

where $\text{Exp}_\psi(\cdot)$ is the FS exponential map. In practice, we may replace Exp with a first-order retraction (followed by normalization and phase quotient) for efficiency; both yield similar results at sufficiently small step sizes.

---

**Algorithm 1** Stochastic Schrödinger Diffusion Models (SS-DMs)

---

**Require:** Data samples $\{\psi_0^{(i)}\}$; diffusion horizon $T$; step sizes $\Delta t, \delta t$; schedules $\sigma(t), \lambda(t)$; prior sampler for $p_T$
1: **Training:**
2: **for** each minibatch **do**
3:     Sample $\psi_0 \sim p_0$ and $t \sim \mathcal{U}(\delta t, T)$
4:     Simulate the forward Riemannian OU diffusion (11) to obtain $(\psi_{t-\delta t}, \psi_t)$
5:     Compute normal coordinates $z \leftarrow \text{log}_{\psi_{t-\delta t}}(\psi_t)$
6:     Compute teacher score $s^{(\text{teach})}(\psi_t, \psi_{t-\delta t}, t, \delta t)$ via Eqs. (16)–(17)
7:     Update $\theta$ by minimizing $\mathcal{L}(\theta)$ in Eq. (19)
8: **end for**
9: **Sampling:**
10: Sample $\psi_T \sim p_T$
11: **for** $k = K, K-1, \ldots, 1$ **do**     ▷ integrate from $T$ to 0
12:     Compute $\tau_k \approx \sigma(t_k)^2 \Delta t$
13:     Draw $\xi_k \sim \mathcal{N}(0, I)$ in $T_{\psi_{t_k}} \mathcal{M}$
14:     $v_k \leftarrow (b(\psi_{t_k}, t_k) - \sigma(t_k)^2 s_\theta(\psi_{t_k}, t_k)) \Delta t + \sqrt{\tau_k} \xi_k$
15:     $\psi_{t_{k-1}} \leftarrow \text{Exp}_{\psi_{t_k}}(v_k)$     ▷ or a retraction
16: **end for**
17: **return** $\psi_0$ as a generated sample from the learned pure-state ensemble

---

## 4. Related Works

**Score-based diffusion models.** Diffusion and score-based generative models learn to sample by reversing a gradual noising process, typically via a learned score $\nabla \log p_t$ and reverse-time SDE/ODE integration (Ho et al., 2020; Song et al., 2021). Extensions beyond Euclidean spaces, including manifolds and other structured domains, replace Euclidean gradients and noise with Riemannian counterparts, often by performing score matching in local normal coordinates using tangent-space perturbations (Bortoli et al., 2022; Huang et al., 2022). Our work follows this geometric line but goes beyond a direct instantiation of Riemannian score-based models on $\mathbb{CP}^{d-1}$. In particular, the strong curvature of the pure-state manifold and the absence of tractable transition densities make standard local score-matching constructions insufficient, motivating a principled local-time training objective and a carefully designed forward diffusion that control geometric bias.

**Quantum generative modeling and diffusion.** Existing quantum generative models, including Born machines (Liu & Wang, 2018; Benedetti et al., 2019; Coyle et al., 2020), quantum Boltzmann machines (Kieferová & Wiebe, 2016; Amin et al., 2018; Zoufal et al., 2020), and quantum GANs (Lloyd & Weedbrook, 2018a; Dallaire-Demers & Killoran, 2018; Zoufal et al., 2019), typically rely on variational circuits trained with adversarial or likelihood-based objectives,

focusing on parameterizing families of quantum circuits. More recent "quantum diffusion" approaches define forward noising via completely positive trace-preserving channels (Chen et al., 2024; Zhu et al., 2025) and learn reverse maps for state recovery or preparation, often using non-unitary circuit constructions such as ancillas and partial trace (Parigi et al., 2025). In contrast, SSDMs adopt a diffusion-based perspective that directly models distributions over quantum pure states on the projective manifold and perform reverse-time sampling via a learned (Riemannian) score field, rather than learning explicit inverse channels or circuit parameterizations. Prior works (Zhang et al., 2023; Kwun et al., 2025) also study generating ensembles of pure states via iterative denoising procedures based on randomization and measurement operations. SSDMs complement these approaches by formulating an intrinsic manifold diffusion with a principled reverse-time sampler and a local-time training signal derived from FS normal-coordinate approximations.

**Stochastic quantum trajectories and unravelings.** Diffusion-like dynamics also appear in measurement-induced trajectories (Dalibard et al., 1992; Gisin & Percival, 1992) and Lindblad unravelings (Kleinekathöfer et al., 2002; Caiaffa et al., 2017; Chen & Kuo, 2025), where stochastic Schrödinger equations describe pure-state paths and motivate recovery/control viewpoints (Kiefer, 2010). SSDMs leverage this connection through an SSE realization, but target a generative modeling objective: learning a score field on $\mathbb{CP}^{d-1}$ and using reverse-time integration to sample from a target ensemble.

**Hybrid pipelines with quantum denoisers.** A separate direction inserts quantum neural components into otherwise classical diffusion models, e.g., quantum neural network (QNN) denoisers (Kölle et al., 2024) for image/latent diffusion (Falco et al., 2024; 2025) and scientific data generation, such as quark and gluon jet synthesis (Baidachna et al., 2025). Our setting differs in that the diffusion itself evolves *quantum states* and the score is defined intrinsically on $\mathbb{CP}^{d-1}$.

## 5. Experiments

We evaluate SSDMs on generative modeling tasks over quantum pure-state ensembles. Our experiments are designed to answer the following questions:

- **RQ1 (Generative Quality).** Can SSDMs generate samples that match the target pure-state ensemble distribution?

- **RQ2 (Geometry Matters).** Does modeling diffusion and scores intrinsically on $\mathbb{CP}^{d-1}$ with the Fubini–Study (FS) geometry improve performance over Euclidean baselines?

- **RQ3 (Local-Time Supervision).** Are local-time analytic teacher scores derived from the FS normal-coordinate OU approximation sufficient for learning a global score field that supports high-quality reverse-time sampling?

- **RQ4 (QML Utility).** Do SSDM-generated quantum encodings improve downstream QML performance via representation-level data augmentation?

Experimental details can be found in Appendix G.

### 5.1. Main Results: Generative Modeling of Pure-State Ensembles

We first evaluate SSDMs on pure-state ensemble generation for $n = 2, 4$, and 6 qubits. Target ensembles are induced by (i) synthetic quantum feature maps with controlled structure and (ii) quantum encodings of real-world datasets. For each setting, we sample $N$ generated states and evaluate distributional similarity using the metrics described in Section G.1. Table 1 summarizes the main generative performance across different system sizes, while Table 2 reports training-time comparisons, where QDDPM incurs substantially higher cost due to its endpoint-alignment training that requires traversing the full diffusion time horizon during optimization.

**Distribution and fidelity matching.** As shown in Table 1, SSDMs consistently outperform competing baselines across all qubit sizes. In particular, SSDMs achieve the highest fidelity $F_0$ while simultaneously attaining substantially lower MMD and observable mismatch $\Delta_{\mathrm{obs}}$, indicating accurate reproduction of both global and observable-level ensemble statistics. Compared to Euclidean VP-SDE and Riemannian score-based generative models (RSGM), the improvement becomes more pronounced as the system dimension increases, highlighting the benefit of modeling diffusion intrinsically on the pure-state manifold. SSDMs also yield improved entanglement Wasserstein distance, suggesting that the learned Riemannian score captures nonlocal correlations present in the target ensembles rather than only marginal statistics.

### 5.2. Does Geometry Matter?

To isolate the role of geometry (RQ2), we compare intrinsic SSDMs with Euclidean score baselines on $n = 2, 4, 6$ qubits. Table 1 and summarizes the effect of FS geometry.

Respecting FS geometry yields consistent improvements in both distribution matching and entanglement statistics, confirming that learning tangent scores intrinsically on $\mathbb{CP}^{d-1}$ is crucial for stable and accurate sampling.

*Table 1.* Main results on $n = 2, 4, 6$ **qubits**. Higher is better for fidelity $F_0$, while lower is better for MMD, $\Delta_{\mathrm{obs}}$, and entanglement Wasserstein distance.

| $n = 2$ | $F_0 \uparrow$ | MMD $\downarrow$ | $\Delta_{\mathrm{obs}} \downarrow$ | Ent. $W_1 \downarrow$ |
|---|---|---|---|---|
| QDDPM (Zhang et al., 2024) | 0.8991 | $1.3092 \times 10^{-2}$ | $1.0294 \times 10^{-1}$ | $6.9427 \times 10^{-2}$ |
| QGAN (Lloyd & Weedbrook, 2018b) | 0.5261 | $4.3099 \times 10^{-1}$ | $6.0502 \times 10^{-1}$ | $3.4518 \times 10^{-1}$ |
| Euclidean VP-SDE (Song et al., 2021) | 0.2487 | $7.1676 \times 10^{-1}$ | $9.7489 \times 10^{-1}$ | $2.8790 \times 10^{-1}$ |
| RSGM (Bortoli et al., 2022) | 0.2889 | $6.3424 \times 10^{-1}$ | $9.1988 \times 10^{-1}$ | $3.0743 \times 10^{-1}$ |
| **SSDM (ours)** | **0.9582** | $2.0351 \times 10^{-3}$ | $2.8584 \times 10^{-2}$ | $2.6872 \times 10^{-2}$ |
| $n = 4$ | $F_0 \uparrow$ | MMD $\downarrow$ | $\Delta_{\mathrm{obs}} \downarrow$ | Ent. $W_1 \downarrow$ |
| QDDPM (Zhang et al., 2024) | 0.5062 | $2.2183 \times 10^{-1}$ | $4.4425 \times 10^{-1}$ | $4.6388 \times 10^{-1}$ |
| QGAN (Lloyd & Weedbrook, 2018b) | 0.0969 | $7.5063 \times 10^{-1}$ | $8.6419 \times 10^{-1}$ | $6.8635 \times 10^{-1}$ |
| Euclidean VP-SDE (Song et al., 2021) | 0.0539 | $7.7408 \times 10^{-1}$ | $9.0754 \times 10^{-1}$ | $6.6624 \times 10^{-1}$ |
| RSGM (Bortoli et al., 2022) | 0.0859 | $7.0681 \times 10^{-1}$ | $8.6678 \times 10^{-1}$ | $6.6605 \times 10^{-1}$ |
| **SSDM (ours)** | **0.8144** | $1.4391 \times 10^{-2}$ | $9.7529 \times 10^{-2}$ | $1.6561 \times 10^{-1}$ |
| $n = 6$ | $F_0 \uparrow$ | MMD $\downarrow$ | $\Delta_{\mathrm{obs}} \downarrow$ | Ent. $W_1 \downarrow$ |
| QDDPM (Zhang et al., 2024) | 0.1435 | $3.9985 \times 10^{-1}$ | $5.0898 \times 10^{-1}$ | $1.9413 \times 10^{-1}$ |
| QGAN (Lloyd & Weedbrook, 2018b) | 0.0246 | $4.7364 \times 10^{-1}$ | $6.7442 \times 10^{-1}$ | $6.8376 \times 10^{-1}$ |
| Euclidean VP-SDE (Song et al., 2021) | 0.0143 | $4.6681 \times 10^{-1}$ | $6.8395 \times 10^{-1}$ | $6.8180 \times 10^{-1}$ |
| RSGM (Bortoli et al., 2022) | 0.0603 | $4.0124 \times 10^{-1}$ | $6.3683 \times 10^{-1}$ | $6.8753 \times 10^{-1}$ |
| **SSDM (ours)** | **0.3922** | $9.6945 \times 10^{-2}$ | $3.0519 \times 10^{-1}$ | $4.6613 \times 10^{-1}$ |

*Table 2.* Training time comparison on $n = 6$ qubits. Lower is better.

| Method | Train time (s) $\downarrow$ | Speedup |
|---|---|---|
| QDDPM | 21150.52 | $1.0\times$ |
| QGAN | 198.64 | $106.4\times$ |
| **SSDM (ours)** | 240.91 | $87.8\times$ |

### 5.3. Local-Time Teacher Ablations

We evaluate the importance of local-time analytic teacher supervision (RQ3) on $n = 2, 4, 6$ qubits. Table 3 and Figure 2 in Appendix G reports the main ablations. Analytic local-time OU teacher scores provide stable and effective supervision. Compared to finite-difference approximations, the analytic teacher reduces estimation noise and improves reverse-time sampling quality.

### 5.4. QML Application: Representation-Level Data Augmentation

We evaluate SSDMs for representation-level augmentation in quantum feature space (RQ4). Given a labeled dataset $\{(x_i, y_i)\}$, we encode inputs into quantum feature states $|\psi(x_i)\rangle$ and train a downstream model using either: (i) the original training set, or (ii) an augmented set that includes additional SSDM-generated states. We train kernel SVM (or ridge regression) using the overlap kernel $k(x, x') = |\langle \psi(x) | \psi(x') \rangle|^2$. We report test accuracy and kernel alignment.

We also evaluate a variational classifier trained on measured features of $|\psi(x)\rangle$, and test whether SSDM augmentation improves generalization under limited labeled data. Table 4 in Appendix G summarizes downstream performance. SSDM-generated states improve downstream performance under limited labeled data, supporting the usefulness of generative modeling for representation-level augmentation.

## 6. Conclusion

We proposed *Stochastic Schrödinger Diffusion Models* (SSDMs), a score-based generative framework for modeling quantum pure-state ensembles from an ensemble-level perspective. By formulating diffusion intrinsically on the complex projective manifold $\mathbb{CP}^{d-1}$ endowed with the Fubini–Study geometry, SSDMs extend classical diffusion models to the quantum setting in a principled and geometry-aware manner. Experiments demonstrate accurate ensemble generation and consistent improvements in downstream quantum machine learning tasks via representation-level data augmentation. More broadly, SSDMs highlight a practical trade-off between physical quantum state preparation and statistical ensemble modeling, enabling efficient representation-level sampling via classical generative surrogates. This perspective is especially relevant in near-term regimes with limited quantum resources, where classical generative models can complement QML workflows before large-scale fault-tolerant devices become available.

## Impact Statement

This paper presents work whose goal is to advance the field of machine learning. There are many potential societal consequences of our work, none of which we feel must be specifically highlighted here.

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

## A. Induced Manifold Diffusion from Tangent-Projected Stratonovich Dynamics

We formalize the statement that a tangent-projected Stratonovich dynamics on the Hilbert sphere induces a diffusion on $\mathbb{CP}^{d-1}$ whose generator matches the intrinsic FS diffusion up to explicit connection terms.

**Proposition A.1** (Induced diffusion and generator on $\mathbb{CP}^{d-1}$). *Let $\pi : \mathbb{S}^{2d-1} \to \mathbb{CP}^{d-1}$ be the quotient map under the $U(1)$ action, and let $\mathcal{H}_\psi = \{u \in T_\psi \mathbb{S}^{2d-1} : \langle \psi, u \rangle = 0\}$ be the horizontal distribution (FS connection), so that $\pi$ is a Riemannian submersion and $\pi_* : \mathcal{H}_\psi \to T_{[\psi]} \mathbb{CP}^{d-1}$ is an isomorphism.*

*Consider the Stratonovich SDE on $\mathbb{S}^{2d-1}$:*

$$d\psi_t = b(\psi_t, t)\, dt + \sigma(t) \sum_{k=1}^{K} V_k(\psi_t) \circ dW_t^{(k)}, \tag{22}$$

*where for each $(\psi, t)$ we have $b(\psi, t) \in \mathcal{H}_\psi$ and $V_k(\psi) \in \mathcal{H}_\psi$, and each $V_k$ is $U(1)$-equivariant so that $e_k([\psi]) := \pi_* V_k(\psi)$ is well-defined on $\mathbb{CP}^{d-1}$.*

*Define $x_t := [\psi_t] \in \mathbb{CP}^{d-1}$ and set*

$$a(x, t) := \pi_* b(\psi, t), \qquad e_k(x) := \pi_* V_k(\psi), \qquad x = [\psi]. \tag{23}$$

*Then $x_t$ satisfies the induced Stratonovich SDE on $\mathbb{CP}^{d-1}$:*

$$dx_t = a(x_t, t)\, dt + \sigma(t) \sum_{k=1}^{K} e_k(x_t) \circ dW_t^{(k)}. \tag{24}$$

*Moreover, for any $f \in C^\infty(\mathbb{CP}^{d-1})$, the generator of $x_t$ is*

$$\mathcal{L}_t f = \langle a(\cdot, t), \nabla_{\mathrm{FS}} f \rangle_{\mathrm{FS}} + \frac{\sigma(t)^2}{2} \sum_{k=1}^{K} \nabla_{e_k} \nabla_{e_k} f, \tag{25}$$

*where $\nabla$ is the Levi–Civita connection of $g_{\mathrm{FS}}$. If in addition $\{e_k(x)\}_{k=1}^{K}$ is an orthonormal frame of $T_x \mathbb{CP}^{d-1}$ (so $K = \dim \mathbb{CP}^{d-1} = 2d - 2$ locally), then*

$$\mathcal{L}_t f = \langle a(\cdot, t), \nabla_{\mathrm{FS}} f \rangle_{\mathrm{FS}} + \frac{\sigma(t)^2}{2} \Delta_{\mathrm{FS}} f + \mathcal{R}_t f, \tag{26}$$

*with an explicit remainder term*

$$\mathcal{R}_t f := \frac{\sigma(t)^2}{2} \sum_{k=1}^{K} \langle \nabla_{\mathrm{FS}} f, \ \nabla_{e_k} e_k \rangle_{\mathrm{FS}}. \tag{27}$$

*In particular, $\mathcal{R}_t f \equiv 0$ at any point $x$ where the chosen orthonormal frame is geodesic (i.e., $\nabla_{e_k} e_k(x) = 0$ for all $k$).*

*Proof.* **Step 1 (Well-defined induced process and induced SDE).** Since $b(\psi, t)$ and $V_k(\psi)$ are horizontal and $U(1)$-equivariant, their pushforwards $a(x, t)$ and $e_k(x)$ in (23) are well-defined on $\mathbb{CP}^{d-1}$. Let $f \in C^\infty(\mathbb{CP}^{d-1})$ and define its lift $\bar{f} := f \circ \pi$ on $\mathbb{S}^{2d-1}$. By the Stratonovich chain rule,

$$d(f(x_t)) = d(\bar{f}(\psi_t)) = \langle \nabla \bar{f}(\psi_t), d\psi_t \rangle = (b\,\bar{f})(\psi_t, t)\, dt + \sigma(t) \sum_k (V_k \bar{f})(\psi_t) \circ dW_t^{(k)}. \tag{28}$$

Using $\bar{f} = f \circ \pi$ and the definition of pushforward, for any horizontal vector field $V$ we have

$$(V \bar{f})(\psi) = (\pi_* V\, f)([\psi]). \tag{29}$$

Applying this to $b$ and $V_k$ turns (28) into

$$d(f(x_t)) = (af)(x_t, t)\, dt + \sigma(t) \sum_k (e_k f)(x_t) \circ dW_t^{(k)}.$$

Since this holds for all smooth test functions $f$, it identifies the induced Stratonovich SDE (24).

**Step 2 (Generator in a chosen frame).** For a Stratonovich SDE on a manifold $dx_t = a\, dt + \sum_k \sigma e_k \circ dW_t^{(k)}$, the generator acting on $f$ is (standard)

$$\mathcal{L}_t f = af + \frac{\sigma(t)^2}{2} \sum_{k=1}^{K} e_k(e_k f). \tag{30}$$

Using the Levi–Civita connection, $e_k(e_k f) = \nabla_{e_k} \nabla_{e_k} f$ for scalar $f$, which gives (25).

**Step 3 (Relation to the Laplace–Beltrami operator and the explicit remainder).** If $\{e_k\}$ is an orthonormal frame, the FS Laplace–Beltrami operator satisfies the local identity

$$\Delta_{\mathrm{FS}} f = \sum_{k=1}^{K} \left( \nabla_{e_k} \nabla_{e_k} f - \nabla_{\nabla_{e_k} e_k} f \right). \tag{31}$$

Rearranging (31) yields

$$\sum_{k=1}^{K} \nabla_{e_k} \nabla_{e_k} f = \Delta_{\mathrm{FS}} f + \sum_{k=1}^{K} \nabla_{\nabla_{e_k} e_k} f.$$

Since $\nabla_v f = \langle \nabla_{\mathrm{FS}} f, v \rangle_{\mathrm{FS}}$ for any vector field $v$, we obtain

$$\sum_{k=1}^{K} \nabla_{\nabla_{e_k} e_k} f = \sum_{k=1}^{K} \langle \nabla_{\mathrm{FS}} f, \ \nabla_{e_k} e_k \rangle_{\mathrm{FS}}.$$

Substituting into (25) gives (26) and the explicit remainder (27). Finally, if the orthonormal frame is geodesic at $x$ (so $\nabla_{e_k} e_k(x) = 0$), then $\mathcal{R}_t f(x) = 0$. $\qquad\square$

## B. SSE Realization and the Induced Diffusion on $\mathbb{CP}^{d-1}$

This appendix explains how the Stratonovich stochastic Schrödinger dynamics on the unit Hilbert sphere induces an (approximately) isotropic diffusion on the projective manifold $\mathbb{CP}^{d-1}$ after quotienting out the global phase. We also clarify in what sense the induced generator matches the intrinsic manifold diffusion in Eq. (8) up to curvature/connection terms.

### B.1.  From the Hilbert Sphere to the Projective Manifold

Let $\mathbb{S}^{2d-1} = \{\psi \in \mathbb{C}^d : \langle \psi, \psi \rangle = 1\}$ be the unit sphere in $\mathbb{C}^d$ equipped with the standard (real) Riemannian structure. The complex projective space $\mathbb{CP}^{d-1}$ is obtained as the quotient $\mathbb{S}^{2d-1}/U(1)$ under the global phase action $\psi \sim e^{i\theta}\psi$, with the canonical projection

$$\pi : \mathbb{S}^{2d-1} \to \mathbb{CP}^{d-1}, \qquad \pi(\psi) = [\psi]. \tag{32}$$

The vertical space at $\psi$ is spanned by the infinitesimal phase direction $v(\psi) = i\psi$. We use the horizontal distribution

$$\mathcal{H}_\psi := \{u \in T_\psi \mathbb{S}^{2d-1} : \ \langle \psi, u \rangle = 0\}, \tag{33}$$

which removes the phase component and is compatible with the Fubini–Study geometry.

**Lemma B.1** (Quotient structure and horizontal lift). *With the horizontal distribution (33), $\pi$ is a Riemannian submersion onto $(\mathbb{CP}^{d-1}, g_{\mathrm{FS}})$. Moreover, for each $\psi \in \mathbb{S}^{2d-1}$, the differential $\pi_*$ restricts to an isomorphism $\mathcal{H}_\psi \cong T_{[\psi]}\mathbb{CP}^{d-1}$.*

**Lemma B.2** (Frame independence under phase). *Let $V(\psi) \in \mathcal{H}_\psi$ be a horizontal vector field on $\mathbb{S}^{2d-1}$ satisfying $U(1)$-equivariance: $V(e^{i\theta}\psi) = e^{i\theta}V(\psi)$. Then the pushforward $e([\psi]) := \pi_* V(\psi)$ is well-defined on $\mathbb{CP}^{d-1}$ (independent of the representative of $[\psi]$).*

### B.2.  Horizontal Projection of SSE Vector Fields

Consider the Stratonovich SSE on $\mathbb{S}^{2d-1}$ (Eq. (10) in the main text):

$$d\,|\psi_t\rangle = -\,iH(t)\,|\psi_t\rangle \ dt \ - \ \frac{1}{2}\eta(t) \sum_k G_k^2 \,|\psi_t\rangle \ dt \ - \ i\sqrt{\eta(t)} \sum_k G_k \,|\psi_t\rangle \circ dW_t^{(k)}. \tag{34}$$

The stochastic term is generated by vector fields $X_k(\psi) := -i\, G_k \psi$ on $\mathbb{S}^{2d-1}$.

In general, $X_k(\psi)$ contains a vertical (phase) component. We define its horizontal projection by removing the component along $\psi$:

$$\widetilde{X}_k(\psi) := X_k(\psi) - \langle \psi, X_k(\psi)\rangle\, \psi. \tag{35}$$

**Lemma B.3** (Horizontal projection removes the global phase component). *For any $X(\psi) \in T_\psi \mathbb{S}^{2d-1}$, define $\widetilde{X}(\psi) := X(\psi) - \langle \psi, X(\psi)\rangle\, \psi$. Then $\widetilde{X}(\psi) \in \mathcal{H}_\psi$. In particular, for $X_k(\psi) = -iG_k\psi$, the induced fields on $\mathbb{CP}^{d-1}$ defined by*

$$e_k([\psi]) := \pi_* \widetilde{X}_k(\psi)\ \in\ T_{[\psi]}\mathbb{CP}^{d-1} \tag{36}$$

*are well-defined.*

### B.3.  Induced Generator on $\mathbb{CP}^{d-1}$

**Proposition B.4** (Induced generator on $\mathbb{CP}^{d-1}$). *Consider a Stratonovich SDE on $\mathbb{S}^{2d-1}$ driven by horizontal vector fields:*

$$d\psi_t = V_0(\psi_t, t)\, dt + \sum_{k=1}^K V_k(\psi_t, t) \circ dW_t^{(k)}, \tag{37}$$

*where $V_k(\psi, t) \in \mathcal{H}_\psi$ and the induced fields $e_k([\psi], t) := \pi_* V_k(\psi, t)$ are well-defined on $\mathbb{CP}^{d-1}$. Then the quotient process $x_t := [\psi_t] \in \mathbb{CP}^{d-1}$ is a diffusion whose generator satisfies, for all $f \in C^\infty(\mathbb{CP}^{d-1})$,*

$$\mathcal{L}_t^{\mathbb{CP}} f = e_0 f + \frac{1}{2} \sum_{k=1}^K e_k(e_k f), \tag{38}$$

*where $e_0$ is the pushforward of the horizontal component of $V_0$.*

*If, in addition, the diffusion directions are (approximately) isotropic under the FS metric, in the sense that*

$$\sum_{k=1}^K \langle u, e_k(x, t)\rangle_{\mathrm{FS}}^2 = \|u\|_{\mathrm{FS}}^2 \quad \textit{for all } u \in T_x \mathbb{CP}^{d-1}, \tag{39}$$

*then*

$$\frac{1}{2} \sum_{k=1}^K e_k(e_k f) = \frac{1}{2}\Delta_{\mathrm{FS}} f + \mathcal{R}_t f, \tag{40}$$

*where $\mathcal{R}_t$ collects Levi–Civita connection terms induced by the (generally non-parallel) local frame $\{e_k\}$. Consequently, the induced generator can be written as*

$$\mathcal{L}_t^{\mathbb{CP}} f = \langle a(\cdot, t), \nabla_{\mathrm{FS}} f\rangle_{\mathrm{FS}} + \frac{\sigma(t)^2}{2}\Delta_{\mathrm{FS}} f + \sigma(t)^2\, \mathcal{R}_t f, \tag{41}$$

*which matches the intrinsic FS diffusion generator in Eq. (8) up to curvature/connection effects.*

**Connection to the SSE in Eq. (10).**  Eq. (34) fits into Proposition B.4 by taking $V_k(\psi, t) = -i\sqrt{\eta(t)}\, \widetilde{X}_k(\psi)$ (and incorporating the remaining deterministic terms into $V_0$). Up to a normalization constant that depends on the convention for $\{G_k\}$, $\sigma(t)^2$ is proportional to $\eta(t)$.

### B.4.  What "Up to Curvature Terms" Means in Practice

The remainder $\mathcal{R}_t$ in Eq. (40) arises because: (i) on a curved manifold, $\Delta_{\mathrm{FS}}$ is the trace of the covariant Hessian, whereas $\sum_k e_k(e_k \cdot)$ depends on the chosen local frame and introduces connection terms; and (ii) the pushed-forward fields $\{e_k\}$ constructed from a fixed Lie-algebra basis $\{G_k\}$ need not coincide with a geodesic orthonormal frame at every point.

**Corollary B.5** (Small-step regime suppresses curvature remainder). *Assume the reverse-time sampler and the local-time objective use a step size $\delta t$ and map tangent increments back to $\mathcal{M}$ via $\mathrm{Exp}$ (or a retraction) in locally orthonormal frames. Then the accumulated effect of $\mathcal{R}_t$ scales to higher order in $\delta t$ (and vanishes as $\delta t \to 0$), so the resulting discrete-time updates approximate the intrinsic diffusion (8) without requiring explicit coordinate correction terms.*

### B.5. Sanity Checks for the SSE-Induced Isotropy

**Proposition B.6** (Soundness of practical isotropy diagnostics). *Consider a drift-free diffusion $(x_t)_{t \geq 0}$ on $(\mathbb{CP}^{d-1}, g_{FS})$ with generator*

$$\mathcal{L}f = \frac{\sigma^2}{2}\Delta_{FS}f, \qquad f \in C^\infty(\mathbb{CP}^{d-1}), \tag{42}$$

*i.e., (time-homogeneous) FS-Brownian motion up to a diffusion-rate factor $\sigma^2$. Then:*

*(i) The unitarily-invariant FS/Haar measure $\mu_{FS}$ is stationary for $(x_t)$, i.e., if $x_0 \sim \mu_{FS}$ then $x_t \sim \mu_{FS}$ for all $t \geq 0$.*

*(ii) (Moment/observable test.) For any bounded measurable observable $\phi : \mathbb{CP}^{d-1} \to \mathbb{R}$,*

$$\mathbb{E}_{x \sim \mu_{FS}}[\phi(x)] = \lim_{t \to \infty} \mathbb{E}\big[\phi(x_t) \mid x_0 = x\big] \quad \text{for } \mu_{FS}\text{-a.e. } x, \tag{43}$$

*whenever the process is ergodic w.r.t. $\mu_{FS}$. In particular, empirical averages of low-order overlap/observable statistics computed from long-time samples converge to the corresponding FS/Haar expectations.*

*(iii) (Generator test.) For any $f \in C^\infty(\mathbb{CP}^{d-1})$,*

$$\mathbb{E}\left[\frac{f(x_{t+\delta}) - f(x_t)}{\delta} \,\middle|\, x_t = x\right] \xrightarrow[\delta \downarrow 0]{} (\mathcal{L}f)(x) = \frac{\sigma^2}{2}(\Delta_{FS}f)(x), \tag{44}$$

*so short-time numerical estimates of the generator on probe functions necessarily scale with $\Delta_{FS}$.*

*Consequently, if an SSE-induced (or numerically implemented) dynamics is a faithful discretization/realization of the isotropic FS diffusion (42), then diagnostics based on (ii)–(iii) must hold. Conversely, passing these diagnostics for a finite family of observables/probe functions provides empirical support but does not by itself imply full isotropy.*

*Proof.* **(i) Stationarity of $\mu_{FS}$.** Let $\mu_{FS}$ denote the Riemannian volume measure induced by $g_{FS}$, normalized to be a probability measure. On a compact boundaryless Riemannian manifold, the Laplace–Beltrami operator is symmetric w.r.t. the volume measure: for all $f, g \in C^\infty(\mathbb{CP}^{d-1})$,

$$\int f \, \Delta_{FS}g \, d\mu_{FS} = \int g \, \Delta_{FS}f \, d\mu_{FS} = -\int \langle \nabla_{FS}f, \nabla_{FS}g \rangle_{FS} \, d\mu_{FS}. \tag{45}$$

In particular, taking $f \equiv 1$ yields $\int \Delta_{FS}g \, d\mu_{FS} = 0$, hence $\int \mathcal{L}g \, d\mu_{FS} = \frac{\sigma^2}{2} \int \Delta_{FS}g \, d\mu_{FS} = 0$. Equivalently, $\mathcal{L}^* \mu_{FS} = 0$, so $\mu_{FS}$ is stationary for the Markov semigroup generated by $\mathcal{L}$.

**(ii) Long-time moment/observable convergence (ergodic case).** Assume ergodicity w.r.t. $\mu_{FS}$ (true for FS-Brownian motion on compact connected manifolds). Then by the ergodic theorem for Markov processes, for any integrable observable $\phi$, time averages (and, under mild additional mixing assumptions, also long-time marginals) converge to $\int \phi \, d\mu_{FS}$. In particular, empirical averages of low-order overlap/observable statistics computed from sufficiently long trajectories converge to the FS/Haar expectations.

**(iii) Generator test.** By definition of the (infinitesimal) generator of a Markov process, for $f$ in the domain of $\mathcal{L}$ (in particular $C^\infty$),

$$(\mathcal{L}f)(x) = \lim_{\delta \downarrow 0} \frac{\mathbb{E}[f(x_{t+\delta}) \mid x_t = x] - f(x)}{\delta}. \tag{46}$$

This gives (44). Substituting (42) yields $(\mathcal{L}f)(x) = \frac{\sigma^2}{2}\Delta_{FS}f(x)$. $\qquad\square$

## C. Time Reversal and Riemannian Score on $\mathbb{CP}^{d-1}$

### C.1. Forward Diffusion Generator

**Proposition C.1** (Forward generator on $(\mathcal{M}, g_{FS})$). *Let $(\mathcal{M}, g_{FS})$ be a Riemannian manifold and consider the time-inhomogeneous diffusion*

$$d\psi_t = b(\psi_t, t) \, dt + \sigma(t) \, dW_t^{(\mathcal{M})}, \tag{47}$$

*where $W_t^{(\mathcal{M})}$ denotes Brownian motion associated with $g_{\mathrm{FS}}$ and $b(\cdot, t)$ is a smooth vector field. Then for any $f \in C^\infty(\mathcal{M})$, the infinitesimal generator of $\psi_t$ is*

$$(\mathcal{L}_t f)(\psi) = \langle b(\psi, t), \nabla_{\mathrm{FS}} f(\psi) \rangle_{\mathrm{FS}} + \frac{\sigma(t)^2}{2} \Delta_{\mathrm{FS}} f(\psi), \tag{48}$$

*where $\nabla_{\mathrm{FS}}$ and $\Delta_{\mathrm{FS}}$ denote the Riemannian gradient and Laplace–Beltrami operator induced by $g_{\mathrm{FS}}$.*

*Proof.* This is the standard generator formula for a diffusion with drift $b$ and isotropic Brownian noise on a Riemannian manifold. The Brownian component contributes $\frac{1}{2}\Delta_{\mathrm{FS}}$, and the scaling by $\sigma(t)$ yields the factor $\frac{\sigma(t)^2}{2}$. □

### C.2. Reverse-Time Dynamics and the Riemannian Score

**Proposition C.2** (Reverse-time drift and Riemannian score). *Let $(\mathcal{M}, g_{\mathrm{FS}})$ be a compact Riemannian manifold without boundary and consider the forward diffusion*

$$d\psi_t = b(\psi_t, t)\, dt + \sigma(t)\, dW_t^{(\mathcal{M})}, \qquad t \in [0, T], \tag{49}$$

*where $W_t^{(\mathcal{M})}$ is Brownian motion associated with $g_{\mathrm{FS}}$. Let $p_t$ denote the density of $\psi_t$ with respect to the Riemannian volume measure, and assume $p_t$ is smooth and strictly positive for $t \in (0, T]$.*

*Then the time-reversed process $\{\psi_{T-t}\}_{t \in [0, T]}$ is again a diffusion on $\mathcal{M}$ with the same diffusion coefficient. In intrinsic Stratonovich form, its dynamics can be written as*

$$d\psi_t = \tilde{b}(\psi_t, t)\, dt + \sigma(t)\, d\bar{W}_t^{(\mathcal{M})}, \tag{50}$$

*where $\bar{W}_t^{(\mathcal{M})}$ is reverse-time Brownian motion and the reverse drift satisfies*

$$\tilde{b}(\psi, t) = b(\psi, t) - \sigma(t)^2\, \nabla_{\mathrm{FS}} \log p_t(\psi). \tag{51}$$

*Equivalently, the reverse drift depends on the* Riemannian score

$$s^\star(\psi, t) := \nabla_{\mathrm{FS}} \log p_t(\psi) \in T_\psi \mathcal{M}. \tag{52}$$

*Proof.* This result follows from the time-reversal theory of nondegenerate diffusions on Riemannian manifolds when the forward diffusion is defined using Brownian motion associated with the Riemannian volume measure. In intrinsic Stratonovich form, the reverse drift differs from the forward drift by $-\sigma(t)^2 \nabla_{\mathrm{FS}} \log p_t$. See, e.g., Haussmann and Pardoux (1986) and Fathi (2021) for rigorous statements. □

## D. Coordinate Form and Itô Corrections for the Reverse-Time SDE

This appendix unpacks the remark in Sec. 3.2 on geometry-dependent correction terms. We state the intrinsic Stratonovich reverse-time SDE and then provide its coordinate/Itô representations.

### D.1. Intrinsic Stratonovich form

Let $(\mathcal{M}, g)$ be a Riemannian manifold and consider the reverse-time diffusion written intrinsically in Stratonovich form

$$d\psi_t = \tilde{b}(\psi_t, t)\, dt + \sigma(t)\, d\bar{W}_t^{(\mathcal{M})}, \tag{53}$$

where $\bar{W}_t^{(\mathcal{M})}$ denotes reverse-time Brownian motion on $(\mathcal{M}, g)$. Equivalently, fixing a (local) orthonormal frame $\{e_i(\cdot)\}_{i=1}^n$ on $\mathcal{M}$ ($n = \dim \mathcal{M}$), one may represent Brownian motion as

$$d\bar{W}_t^{(\mathcal{M})} = \sum_{i=1}^n e_i(\psi_t) \circ d\bar{W}_t^{(i)}, \tag{54}$$

so that (53) becomes

$$d\psi_t = \tilde{b}(\psi_t, t)\, dt + \sigma(t) \sum_{i=1}^n e_i(\psi_t) \circ d\bar{W}_t^{(i)}. \tag{55}$$

### D.2. Conversion to Itô form

Let $\nabla$ be the Levi–Civita connection associated with $g$. The Stratonovich SDE (55) can be converted to an equivalent Itô SDE:

$$d\psi_t = \Big(\tilde{b}(\psi_t, t) + \frac{\sigma(t)^2}{2} \sum_{i=1}^{n} \nabla_{e_i} e_i(\psi_t)\Big) dt + \sigma(t) \sum_{i=1}^{n} e_i(\psi_t) \, d\bar{W}_t^{(i)}. \tag{56}$$

The additional drift term $\frac{\sigma(t)^2}{2} \sum_i \nabla_{e_i} e_i$ is the geometry-dependent Itô–Stratonovich correction; it vanishes at a point where the chosen orthonormal frame is geodesic (normal) (i.e., $\nabla_{e_i} e_i = 0$ at that point).

### D.3. Local coordinate form

Let $(x^1, \ldots, x^n)$ be local coordinates and write the Itô SDE in components:

$$dx_t^\alpha = \Big(\tilde{b}^\alpha(x_t, t) + \frac{\sigma(t)^2}{2} \sum_{i=1}^{n} \big(\nabla_{e_i} e_i\big)^\alpha(x_t)\Big) dt + \sigma(t) \sum_{i=1}^{n} e_i^\alpha(x_t) \, d\bar{W}_t^{(i)}. \tag{57}$$

Equivalently, one may express the correction in terms of Christoffel symbols $\Gamma_{\beta\gamma}^\alpha$ if the diffusion is written using the coordinate basis; such expressions coincide with (56) after identifying $e_i^\alpha$ and using $\nabla_{e_i} e_i^\alpha = e_i^\beta \partial_\beta e_i^\alpha + \Gamma_{\beta\gamma}^\alpha e_i^\beta e_i^\gamma$.

In our implementation, each update is performed in a locally orthonormal frame on $T_\psi \mathcal{M}$ and then mapped back to the manifold using $\mathrm{Exp}$ (or a retraction). For sufficiently small step size $\delta t$, one may choose the frame to be (approximately) normal at the current point, so that $\sum_i \nabla_{e_i} e_i(\psi)$ is $O(\delta t)$ and the induced bias from the Itô–Stratonovich correction is higher order. This is consistent with the small-step regime assumed in our sampler and in the local-time teacher construction.

## E. Local-Time Approximation and Teacher Scores

**Proposition E.1** (Short-time asymptotics and the local teacher score). *Let $(\mathcal{M}, g)$ be a smooth $n$-dimensional Riemannian manifold (in our case, $(\mathbb{CP}^{d-1}, g_{\mathrm{FS}})$ with $n = 2d - 2$). Consider the time-inhomogeneous diffusion*

$$d\psi_t = b(\psi_t, t) \, dt + \sigma(t) \, dW_t^{(\mathcal{M})}, \tag{58}$$

*where $W_t^{(\mathcal{M})}$ is Brownian motion associated with $g$ and $b(\cdot, t)$ is $C^2$ in space and continuous in time. Fix $t \in (0, T]$ and let $\delta t > 0$ be small. Condition on $\psi_{t-\delta t} = \phi$. Assume $\psi_t$ stays within the injectivity radius of $\phi$ (so $\log_\phi$ is well-defined).*

*Let $p_{\delta t}(\psi \mid \phi)$ denote the conditional density of $\psi_t$ given $\phi$, with respect to the Riemannian volume measure. Let $z = \log_\phi(\psi) \in T_\phi \mathcal{M}$ be normal coordinates at $\phi$, and write $r = d_g(\phi, \psi) = \|z\| + O(\|z\|^3)$.*

*Then the short-time transition density admits the heat-kernel asymptotic expansion*

$$p_{\delta t}(\psi \mid \phi) = (2\pi\sigma(t)^2 \delta t)^{-n/2} \exp\Big(-\frac{r^2}{2\sigma(t)^2 \delta t}\Big) J(\phi, \psi)^{-1/2} \Big(1 + O(\delta t)\Big), \tag{59}$$

*where $J(\phi, \psi)$ is the Jacobian determinant of the exponential map (equivalently the volume distortion term) and the $O(\delta t)$ is uniform on compact subsets inside the injectivity neighborhood.*

*Consequently, the intrinsic conditional score satisfies, as $\delta t \downarrow 0$,*

$$\nabla_\psi \log p_{\delta t}(\psi \mid \phi) = -\frac{1}{\sigma(t)^2 \delta t} \nabla_\psi \Big(\frac{r^2}{2}\Big) - \frac{1}{2} \nabla_\psi \log J(\phi, \psi) + O(1), \tag{60}$$

*where $O(1)$ is bounded as $\delta t \downarrow 0$ (locally uniformly away from the cut locus).*

*Moreover, in normal coordinates $z = \log_\phi(\psi)$, the leading term becomes Euclidean:*

$$\big(d \log_\phi\big)_\psi^* \nabla_\psi \log p_{\delta t}(\psi \mid \phi) = -\frac{1}{\sigma(t)^2 \delta t} z + O(\|z\|) + O(1), \tag{61}$$

*so that the Gaussian teacher score $s_z^{(\mathrm{teach})}(z) = -\big(\sigma(t)^2 \delta t\big)^{-1} z$ matches the dominant short-time behavior, and mapping it back via $\big(d \log_\phi\big)_\psi^*$ gives a first-order approximation to the intrinsic conditional score.*

*Proof.* We use standard short-time heat-kernel asymptotics for nondegenerate diffusions on Riemannian manifolds.

**Step 1 (Reduction to a frozen-time operator).** Over a short interval $[t - \delta t, t]$, freeze coefficients at time $t$: the generator is $\mathcal{L}_t = \langle b(\cdot, t), \nabla(\cdot) \rangle + \frac{\sigma(t)^2}{2} \Delta$. The drift contributes only an $O(\delta t)$ shift in the mean and enters the prefactor at order $O(1)$ in the score; the singular $\delta t^{-1}$ behavior is determined by the diffusion (the Laplace–Beltrami part). Therefore, the leading short-time asymptotics are governed by the heat kernel of $\frac{\sigma(t)^2}{2} \Delta$ (with lower-order perturbations absorbed into the $O(\delta t)$ factor in (59)).

**Step 2 (Heat kernel expansion).** Let $q_{\delta t}(\psi, \phi)$ be the heat kernel associated with $\frac{\sigma(t)^2}{2} \Delta$. On the injectivity neighborhood of $\phi$, the classical Minakshisundaram–Pleijel (Hadamard) parametrix gives

$$q_{\delta t}(\psi, \phi) = (2\pi\sigma(t)^2 \delta t)^{-n/2} \exp\Big( -\frac{d_g(\phi, \psi)^2}{2\sigma(t)^2 \delta t} \Big) J(\phi, \psi)^{-1/2} \big( 1 + O(\delta t) \big),$$

uniformly on compact subsets away from the cut locus. This yields (59) for $p_{\delta t}(\psi \mid \phi)$ up to lower-order perturbations induced by $b$, which remain inside the $(1 + O(\delta t))$ factor.

**Step 3 (Differentiate $\log p_{\delta t}$ to obtain the score).** Take $\log$ of (59) and differentiate w.r.t. $\psi$: the normalization term contributes zero gradient, while the exponential term gives

$$\nabla_\psi \log p_{\delta t}(\psi \mid \phi) = -\frac{1}{\sigma(t)^2 \delta t} \nabla_\psi \Big( \frac{d_g(\phi, \psi)^2}{2} \Big) - \frac{1}{2} \nabla_\psi \log J(\phi, \psi) + O(1),$$

where $O(1)$ collects the gradient of $\log(1 + O(\delta t))$ and drift-induced lower-order terms. This proves (60).

**Step 4 (Normal coordinates and leading Euclidean form).** In normal coordinates $z = \log_\phi(\psi)$, we have $\frac{d_g(\phi, \psi)^2}{2} = \frac{\|z\|^2}{2} + O(\|z\|^4)$ and $\big( d \log_\phi \big)^*_\psi \nabla_\psi \frac{d_g(\phi, \psi)^2}{2} = z + O(\|z\|^3)$. Also, $\nabla_\psi \log J(\phi, \psi) = O(\|z\|)$ near $\phi$ (volume distortion starts at quadratic order in normal coordinates). Substituting these into (60) yields (61). $\qquad\square$

# F. Riemannian Denoising Score Matching: Consistency

For fixed $t$ and $\delta t$, the training pair $(\phi, \psi)$ is generated by simulating the forward diffusion starting from data $\psi_0 \sim p_0$, i.e.,

$$(\phi, \psi) = (\psi_{t-\delta t}, \psi_t).$$

This induces a joint density $p(\phi, \psi)$ and the corresponding conditional density $p(\psi \mid \phi)$. The population objective in Eq. (20) is exactly $J[s]$ in Eq. (62) with target

$$u(\psi, \phi) = \nabla_{\mathrm{FS}} \log p(\psi \mid \phi).$$

In practice, we replace $u$ by the local-time teacher approximation in Eqs. (17)–(18), whose consistency in the limit $\delta t \to 0$ is established in Proposition F.2.

**Proposition F.1** (Population optimum of Riemannian DSM equals the marginal score)**.** *Let* $(\mathcal{M}, g_{\mathrm{FS}})$ *be a compact Riemannian manifold without boundary and let* $p_t$ *denote the time-marginal density of the forward diffusion w.r.t. the Riemannian volume measure. Fix* $t \in (0, T]$ *and a step* $\delta t > 0$*. Let* $(\phi, \psi) = (\psi_{t-\delta t}, \psi_t)$ *be drawn from the forward process, and denote the conditional density* $p(\psi \mid \phi)$ *(again w.r.t. the Riemannian volume measure).*

*Consider the population objective over measurable tangent vector fields* $s(\cdot, t) : \mathcal{M} \to T\mathcal{M}$*:*

$$\mathcal{J}[s] := \mathbb{E}\Big[ \lambda(t, \delta t) \, \| s(\psi, t) - \nabla_{\mathrm{FS}} \log p(\psi \mid \phi) \|^2_{\mathrm{FS}} \Big], \tag{62}$$

*where the expectation is over* $\psi_0 \sim p_0$ *and the forward diffusion randomness, and* $\lambda(t, \delta t) > 0$ *is any weight that does not depend on* $s$*.*

*Then any minimizer* $s^\star(\cdot, t)$ *of* (62) *satisfies,* $p_t$*-a.e. in* $\psi$*,*

$$s^\star(\psi, t) = \nabla_{\mathrm{FS}} \log p_t(\psi). \tag{63}$$

*Equivalently, the DSM objective distills conditional scores into the marginal Riemannian score.*

*Proof.* Fix $t$ and abbreviate $S(\psi) := s(\psi, t)$. By conditioning on $\psi$, we can write

$$\mathcal{J}[S] = \mathbb{E}_\psi \Big[ \lambda(t, \delta t) \, \mathbb{E}\big[ \|S(\psi) - U\|_{\mathrm{FS}}^2 \mid \psi \big] \Big],$$

where $U := \nabla_{\mathrm{FS}} \log p(\psi \mid \phi) \in T_\psi \mathcal{M}$. Since $\lambda(t, \delta t) > 0$ is a constant given $(t, \delta t)$, minimization over $S$ is pointwise in $\psi$. For each fixed $\psi$, the unique minimizer of $\mathbb{E}[\|S(\psi) - U\|^2 \mid \psi]$ is

$$S^\star(\psi) = \mathbb{E}[U \mid \psi]. \tag{64}$$

It remains to show $\mathbb{E}[\nabla_{\mathrm{FS}} \log p(\psi \mid \phi) \mid \psi] = \nabla_{\mathrm{FS}} \log p_t(\psi)$. Let $p(\phi, \psi)$ be the joint density of $(\phi, \psi)$ and $p_t(\psi)$ the marginal. Using $p(\psi \mid \phi) = p(\phi, \psi)/p_{t-\delta t}(\phi)$, we have

$$\nabla_{\mathrm{FS}} \log p(\psi \mid \phi) = \nabla_{\mathrm{FS}} \log p(\phi, \psi),$$

because $p_{t-\delta t}(\phi)$ does not depend on $\psi$. Therefore,

$$\mathbb{E}\big[ \nabla_{\mathrm{FS}} \log p(\psi \mid \phi) \mid \psi \big] = \int \nabla_{\mathrm{FS}} \log p(\phi, \psi) \, p(\phi \mid \psi) \, d\phi.$$

Since $p(\phi \mid \psi) = p(\phi, \psi)/p_t(\psi)$, the integral becomes

$$\frac{1}{p_t(\psi)} \int \nabla_{\mathrm{FS}} p(\phi, \psi) \, d\phi = \frac{\nabla_{\mathrm{FS}} p_t(\psi)}{p_t(\psi)} = \nabla_{\mathrm{FS}} \log p_t(\psi),$$

where we used that differentiation w.r.t. $\psi$ commutes with integration in $\phi$ under the stated smoothness/compactness assumptions. Combining with (64) yields (63). $\square$

**Proposition F.2** (Consistency of the local-time teacher objective)**.** *Under the assumptions of Proposition F.1, suppose the teacher score $s^{(\mathrm{teach})}(\psi, \phi, t, \delta t)$ satisfies the short-time approximation*

$$s^{(\mathrm{teach})}(\psi, \phi, t, \delta t) = \nabla_{\mathrm{FS}} \log p(\psi \mid \phi) + \varepsilon(\psi, \phi, t, \delta t), \tag{65}$$

*with*

$$\mathbb{E}\big[ \|\varepsilon(\psi, \phi, t, \delta t)\|_{\mathrm{FS}}^2 \big] \xrightarrow[\delta t \downarrow 0]{} 0, \tag{66}$$

*for each fixed $t$ (uniformly on compact subsets away from the cut locus). Then the population minimizer of the practical objective*

$$\widetilde{\mathcal{J}}[s] := \mathbb{E}\Big[ \lambda(t, \delta t) \, \|s(\psi, t) - s^{(\mathrm{teach})}(\psi, \phi, t, \delta t)\|_{\mathrm{FS}}^2 \Big] \tag{67}$$

*converges (in $L^2(p_t)$) to the marginal score $\nabla_{\mathrm{FS}} \log p_t(\psi)$ as $\delta t \downarrow 0$.*

*Proof.* Let $U := \nabla_{\mathrm{FS}} \log p(\psi \mid \phi)$ and $\hat{U} := s^{(\mathrm{teach})} = U + \varepsilon$. As in Proposition F.1, the pointwise minimizer of (67) satisfies

$$s_{\delta t}^\star(\psi, t) = \mathbb{E}[\hat{U} \mid \psi] = \mathbb{E}[U \mid \psi] + \mathbb{E}[\varepsilon \mid \psi].$$

By Proposition F.1, $\mathbb{E}[U \mid \psi] = \nabla_{\mathrm{FS}} \log p_t(\psi)$. Moreover, by Jensen,

$$\mathbb{E}\big[ \|\mathbb{E}[\varepsilon \mid \psi]\|_{\mathrm{FS}}^2 \big] \leq \mathbb{E}\big[ \|\varepsilon\|_{\mathrm{FS}}^2 \big] \xrightarrow[\delta t \downarrow 0]{} 0,$$

which implies $s_{\delta t}^\star(\cdot, t) \to \nabla_{\mathrm{FS}} \log p_t(\cdot)$ in $L^2(p_t)$. $\square$

**Lemma F.3** (Projection to $T_\psi \mathcal{M}$ does not change the optimum)**.** *Let $u(\psi) \in T_\psi \mathcal{M}$ be any target tangent field and let $\hat{s}(\psi)$ be an arbitrary ambient vector. Then*

$$\|\mathcal{P}_\psi(\hat{s}(\psi)) - u(\psi)\|_{\mathrm{FS}}^2 = \|\mathcal{P}_\psi(\hat{s}(\psi)) - u(\psi)\|_{\mathrm{FS}}^2 + \text{(term independent of $u$)},$$

*and the minimizer over ambient $\hat{s}$ is achieved when $\mathcal{P}_\psi(\hat{s}(\psi)) = u(\psi)$. In particular, including $\mathcal{P}_\psi$ in the loss enforces tangency without altering the target optimum.*

# G. Experimental Setup

**Data: ensembles of quantum feature states.**    We consider pure-state ensembles $\{|\psi(x)\rangle\}$ induced by quantum feature maps applied to classical data $x \sim p_{\mathrm{data}}(x)$. Unless stated otherwise, feature maps are implemented by shallow parameterized quantum circuits and encode inputs into $n$-qubit states $|\psi(x)\rangle \in \mathbb{C}^{2^n}$. We evaluate both synthetic ensembles (for controlled evaluation) and ensembles induced by real-world datasets encoded into quantum feature states.

**Forward diffusion and prior.**    We construct an intrinsic *Riemannian OU* forward diffusion on $\mathbb{CP}^{d-1}$ under the FS metric, with diffusion horizon $T$ and diffusion schedule $\sigma(t)$. For the prior $p_T$, we use a Clifford $t$-design ensemble as the default due to its efficient sampling.

**Score model and training.**    Unless otherwise noted, we parameterize the score model $s_\theta(\psi, t)$ using a classical neural network that outputs a tangent vector field. We train $s_\theta$ using the Riemannian denoising score matching objective in Eq. (19), with analytic local-time teacher scores derived from the FS normal-coordinate OU approximation (Section 3.3).

**Numerical integration and implementation.**    We simulate forward and reverse processes using a manifold-adapted Euler–Maruyama scheme: we take Euler steps in the tangent space and retract back to $\mathbb{CP}^{d-1}$ via the FS exponential map (or a first-order retraction) as in Eq. (21). We report results averaged over $M$ random seeds. All experiments are conducted on an NVIDIA A6000 GPU.

**Reporting protocol.**    In the main paper we report results for $n = 2, 4, 6$ qubits as the largest setting in our evaluation, and defer results for $n \in \{2, 4, 6\}$ . All methods use the same feature map, training budget, and evaluation metrics unless stated otherwise.

## G.1. Evaluation Metrics

Evaluating quantum generative models requires distributional metrics beyond pointwise fidelity. We use a combination of the following:

**Fidelity to target ensemble.**    We additionally report the average state fidelity between generated samples and the target ensemble, defined as

$$F_0 = \mathbb{E}_{\psi \sim p_{\mathrm{gen}}, \, \phi \sim p_{\mathrm{data}}} \left[ \, | \langle \psi | \phi \rangle |^2 \, \right]. \tag{68}$$

Higher values of $F_0$ indicate closer overlap between generated and target pure states.

**Observable statistics.**    Given a set of observables $\{O_j\}_{j=1}^J$, we compare the generated and target ensembles via moment matching:

$$
\begin{aligned}
&\Delta_{\mathrm{obs}} \\
&= \frac{1}{J} \sum_{j=1}^J \left| \mathbb{E}_{\psi \sim p_{\mathrm{gen}}} \left[ \langle \psi | O_j | \psi \rangle \right] - \mathbb{E}_{\psi \sim p_{\mathrm{data}}} \left[ \langle \psi | O_j | \psi \rangle \right] \right|.
\end{aligned}
\tag{69}
$$

**Kernel MMD on pure states.**    We measure distributional similarity using the maximum mean discrepancy (MMD) with an overlap kernel:

$$
\begin{aligned}
k(\psi, \phi) &= | \langle \psi | \phi \rangle |^2, \\
\mathrm{MMD}^2(p, q) &= \mathbb{E}_{p,p}[k] + \mathbb{E}_{q,q}[k] - 2\mathbb{E}_{p,q}[k].
\end{aligned}
\tag{70}
$$

**Entanglement statistics.**    To capture nonlocal structure, we compare distributions of subsystem entanglement, measured by von Neumann entropy:

$$S_A(\psi) = -\mathrm{Tr}\big(\rho_A \log \rho_A\big), \qquad \rho_A = \mathrm{Tr}_{\bar{A}} |\psi\rangle \langle\psi|. \tag{71}$$

We report Wasserstein distances between entropy histograms and compare mean/variance. To compare entanglement statistics at the distributional level, we compute the entropic Wasserstein-1 distance between histograms of subsystem entropies:

$$\mathrm{Ent.} \, \mathrm{W}_1(p, q) = W_1^\varepsilon\big(\mathcal{H}_p(S_A), \mathcal{H}_q(S_A)\big), \tag{72}$$

where $\mathcal{H}(\cdot)$ denotes the empirical entropy histogram and $W_1^\varepsilon$ is the entropy-regularized Wasserstein distance. Lower values indicate closer agreement of entanglement structure.

**Downstream QML performance.** For QML utility, we evaluate classification or regression performance using (i) quantum kernel methods and (ii) variational quantum classifiers, trained on original and augmented quantum encodings (Section 5.4).

### G.2. Baselines

We compare SSDMs with the following baselines:

- **Euclidean VP-SDE.** A naive score-based diffusion model treating normalized quantum states as Euclidean vectors in $\mathbb{C}^d$.

- **QGAN** (Lloyd & Weedbrook, 2018b). A variational quantum generative adversarial model trained via measurement-based losses.

- **QuDDPM** (Zhang et al., 2024). A circuit-based quantum denoising diffusion probabilistic model.

- **RSGM** (Bortoli et al., 2022). A Riemannian score-based diffusion model on the manifold of pure quantum states. We adapt RSGM to $\mathrm{CP}^{d-1}$ using the standard Riemannian Brownian motion construction, without local-time analytic teacher supervision.

- **Ablated SSDMs.** Variants of our method with key components removed.

In addition to generative quality, we evaluate the representation-level impact of SSDM-based augmentation on downstream quantum machine learning tasks.

**Kernel alignment.** Kernel alignment measures the agreement between the task kernel and label similarity. Given a kernel matrix $K$ and label vector $y$, the alignment is defined as

$$\mathrm{KA}(K, y) = \frac{\langle K, yy^\top \rangle_F}{\|K\|_F \, \|yy^\top\|_F}, \tag{73}$$

where $\langle \cdot, \cdot \rangle_F$ denotes the Frobenius inner product. Higher kernel alignment indicates a more task-relevant quantum representation.

**Kernel gap.** The kernel gap quantifies class separability in feature space by measuring the difference between intra-class and inter-class similarities:

$$\mathrm{Gap}(K) = \mathbb{E}_{(i,j) \in \mathcal{S}}[K_{ij}] - \mathbb{E}_{(i,j) \in \mathcal{D}}[K_{ij}], \tag{74}$$

where $\mathcal{S}$ and $\mathcal{D}$ denote sets of same-class and different-class pairs, respectively. Larger kernel gaps correspond to improved class separation.

**Mean margin.** We further evaluate the margin induced by the quantum kernel classifier. Given the learned decision function $f(\cdot)$, the margin of a sample $(x_i, y_i)$ is

$$\gamma_i = y_i f(x_i), \tag{75}$$

and we report the mean margin over the training set:

$$\mathrm{Margin} = \frac{1}{N} \sum_{i=1}^N \gamma_i. \tag{76}$$

Higher mean margins indicate more robust and discriminative representations.

These representation-level metrics complement generative evaluation by assessing how improved sample quality translates into more discriminative and task-aligned quantum representations.

All baselines are evaluated under the same statevector access assumption and classical simulation setting. Circuit-based methods are simulated classically, and no hardware noise is assumed.

*Table 3.* Ablations on local-time supervision and score construction for $n = 2, 4, 6$ **qubits** (RQ3). Higher is better for fidelity $F_0$, while lower is better for MMD, $\Delta_{\text{obs}}$, and entanglement Wasserstein distance.

| $n = 2$ | $F_0 \uparrow$ | MMD $\downarrow$ | $\Delta_{\text{obs}} \downarrow$ | Ent. $W_1 \downarrow$ |
|---|---|---|---|---|
| No local teacher | 0.6818 | $1.1912 \times 10^{-1}$ | $3.9591 \times 10^{-1}$ | $2.1341 \times 10^{-1}$ |
| Finite-difference teacher | 0.9381 | $2.3944 \times 10^{-3}$ | $5.4271 \times 10^{-2}$ | $4.8452 \times 10^{-2}$ |
| Analytic OU teacher (ours) | **0.9582** | $2.0351 \times 10^{-3}$ | $2.8584 \times 10^{-2}$ | $2.6872 \times 10^{-2}$ |

| $n = 4$ | $F_0 \uparrow$ | MMD $\downarrow$ | $\Delta_{\text{obs}} \downarrow$ | Ent. $W_1 \downarrow$ |
|---|---|---|---|---|
| No local teacher | 0.1579 | $5.8873 \times 10^{-1}$ | $7.8866 \times 10^{-1}$ | $6.5886 \times 10^{-1}$ |
| Finite-difference teacher | 0.6763 | $6.1424 \times 10^{-2}$ | $2.3786 \times 10^{-1}$ | $3.3521 \times 10^{-1}$ |
| Analytic OU teacher (ours) | **0.8144** | $1.4391 \times 10^{-2}$ | $9.7529 \times 10^{-2}$ | $1.6561 \times 10^{-1}$ |

| $n = 6$ | $F_0 \uparrow$ | MMD $\downarrow$ | $\Delta_{\text{obs}} \downarrow$ | Ent. $W_1 \downarrow$ |
|---|---|---|---|---|
| No local teacher | 0.0213 | $4.5677 \times 10^{-1}$ | $6.8235 \times 10^{-1}$ | $6.9585 \times 10^{-1}$ |
| Finite-difference teacher | 0.1476 | $2.9737 \times 10^{-1}$ | $5.5325 \times 10^{-1}$ | $6.5607 \times 10^{-1}$ |
| Analytic OU teacher (ours) | **0.3922** | $9.6945 \times 10^{-2}$ | $3.0519 \times 10^{-1}$ | $4.6613 \times 10^{-1}$ |

*Table 4.* Representation-level improvements induced by SSDM-based augmentation in downstream QML tasks (RQ4). Results are reported for $n = 2, 4, 6$ qubits.

| Qubits | Training Set | Kernel Alignment $\uparrow$ | Kernel Gap $\uparrow$ | Mean Margin $\uparrow$ |
|---|---|---|---|---|
| $n = 2$ | Original only | 0.1749 | 0.0155 | 0.0304 |
| | + SSDM augmentation | **0.2747** | **0.0254** | **0.0527** |
| $n = 4$ | Original only | 0.1054 | 0.0109 | 0.0126 |
| | + SSDM augmentation | **0.1594** | **0.0201** | 0.0118 |
| $n = 6$ | Original only | 0.0831 | 0.0039 | 0.0559 |
| | + SSDM augmentation | **0.1053** | **0.0081** | 0.0518 |

# H. Discussion

We proposed *Stochastic Schrödinger Diffusion Models* (SSDMs), a score-based generative framework for learning and sampling distributions over quantum pure states on $\mathbb{CP}^{d-1}$. SSDMs generalize Euclidean score-based diffusion to the quantum setting by (i) designing an intrinsic *Riemannian OU* forward diffusion on the pure-state manifold with a stochastic Schrödinger (SSE) realization, and (ii) learning a *Riemannian score field* under the Fubini–Study (FS) geometry. A key practical challenge—the intractability of transition densities on curved manifolds—is addressed via a *local-time* learning signal derived from the *local Euclidean OU limit* of the forward diffusion in FS normal coordinates. This yields an analytic Gaussian teacher score for short-time transitions in the tangent space, which is mapped back to the manifold via the logarithm map. In FS normal coordinates, the forward diffusion reduces to a first-order OU/VP process with curvature corrections, providing a principled bridge to classical score-based diffusion models.

## H.1. Relation to Prior Work and the Role of Generative Sampling

Existing quantum diffusion efforts often emphasize (a) channel-based diffusion and learned inverse maps, or (b) stochastic trajectory formulations with recovery/control interpretations. In contrast, SSDMs directly target *generative modeling* of *pure-state ensembles* by enabling *sampling* from an implicit distribution $p_0(\psi)$. Our objective aligns naturally with QML, where data are frequently encoded as pure states and downstream tasks may benefit from learning and generating quantum representations.

A conceptual difference is that SSDMs treat the state space as a Riemannian manifold and formulate the reverse-time dynamics in terms of the *Riemannian score*:

$$s^{\star}(\psi, t) = \nabla_{\text{FS}} \log p_t(\psi) \in T_{\psi} \mathbb{CP}^{d-1}. \tag{77}$$

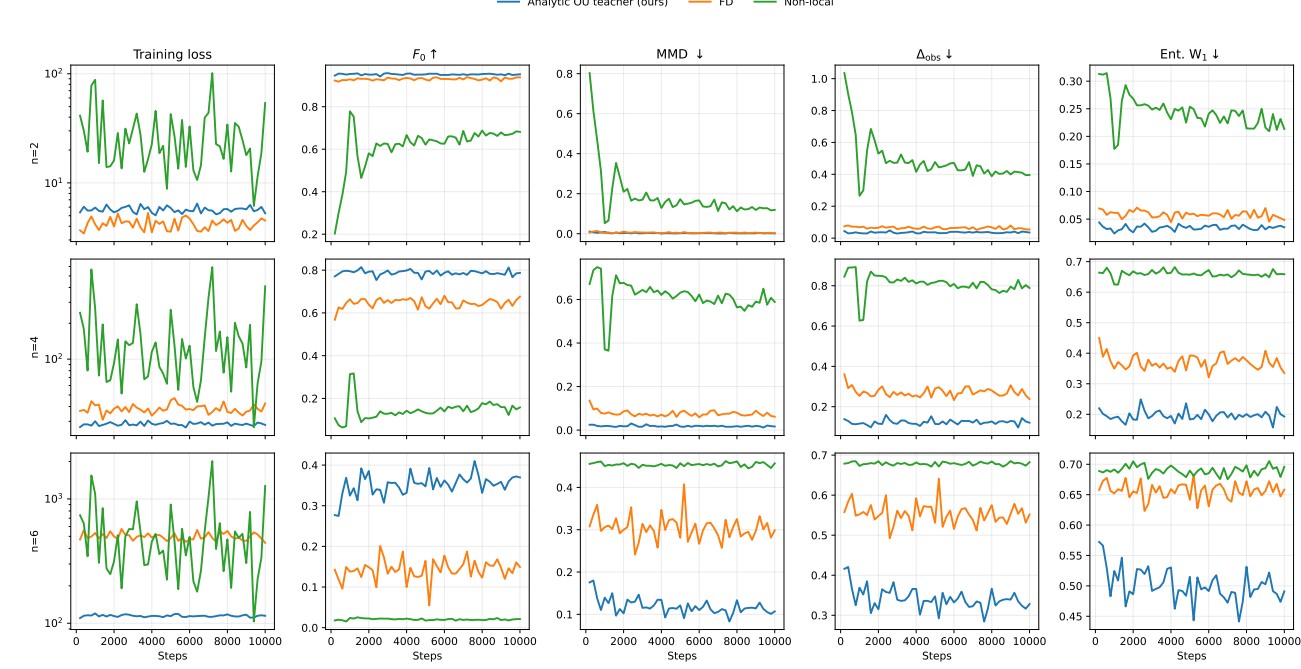

*Figure 2.* Training stability across system sizes $n \in \{2, 4, 6\}$ (rows). Columns report training loss, $F_0$, MMD, $\Delta_{\text{obs}}$, and entropic Wasserstein distance (Ent. $W_1$).

This geometric viewpoint yields a coordinate-free generalization of the score-based paradigm and avoids gauge ambiguities associated with global phase.

## H.2. Why Stochastic Schrödinger Diffusion?

Beyond providing a physical realization, the SSE perspective clarifies how to design forward diffusions that remain stable on curved pure-state manifolds while retaining compatibility with score-based time reversal. SSE-based dynamics naturally define stochastic *pure-state* trajectories while remaining consistent with open-system evolution in expectation. Beyond physical motivation, SSEs provide a flexible *design principle* for constructing stable forward diffusions on $\mathbb{CP}^{d-1}$. In SSDMs, we make this precise by designing a *Riemannian OU* diffusion on the pure-state manifold and providing an SSE realization, thereby obtaining an intrinsic forward process that is OU/VP to first order in FS normal coordinates.

The core quantity required for sampling appears explicitly in the reverse-time drift. For the intrinsic manifold diffusion

$$d\psi_t = b(\psi_t, t)\, dt + \sigma(t)\, dW_t^{(\mathcal{M})}, \tag{78}$$

time reversal yields the reverse-time sampler

$$
\begin{aligned}
d\psi_t &= \big(b(\psi_t, t) - \sigma(t)^2\, s^\star(\psi_t, t)\big)\, dt + \sigma(t)\, d\bar{W}_t^{(\mathcal{M})}, \\
s^\star(\psi, t) &= \nabla_{\text{FS}} \log p_t(\psi),
\end{aligned}
\tag{79}
$$

up to coordinate-form geometry corrections (cf. Appendix D). Eq. (79) mirrors the Euclidean reverse SDE, but with the *Riemannian score* under the FS metric. Once a model $s_\theta(\psi, t) \approx s^\star(\psi, t)$ is learned, sampling becomes straightforward by integrating Eq. (79) from a tractable prior $p_T$.

## H.3. Learning Signals via Local OU Normal Coordinates

A practical limitation in manifold diffusion modeling is that the marginal density $p_t(\psi)$ (and hence the Riemannian score $s^\star(\psi, t) = \nabla_{\text{FS}} \log p_t(\psi)$) is not available in closed form on $\mathbb{CP}^{d-1}$. Rather than relying on explicit transition-kernel

formulas, SSDMs leverage a *local-time* supervision signal based on a geometric fact: in FS normal coordinates, the intrinsic manifold diffusion admits a *local Euclidean OU/VP approximation* to first order, with curvature and volume-element effects entering only at higher order.

**Analytic local-time teacher score.**   Given a short step size $\delta t > 0$, we form local-time pairs $(\phi, \psi) := (\psi_{t-\delta t}, \psi_t)$ from the forward diffusion and define FS normal coordinates centered at $\phi$ by

$$z := \log_\phi(\psi) \in T_\phi\mathcal{M}. \tag{80}$$

In these coordinates, the short-time transition is well-approximated by a Euclidean OU/VP step

$$z \;\approx\; \beta(t, \delta t)\,\xi, \qquad \xi \sim \mathcal{N}(0, I), \tag{81}$$

where $\beta(t, \delta t)$ is determined by the diffusion schedule (with curvature corrections entering at higher order). Consequently, the induced local conditional density $q(z \mid 0)$ is Gaussian, yielding a closed-form teacher score in the tangent space:

$$s_z^{(\text{teach})}(z, t, \delta t) := \nabla_z \log q(z \mid 0) = -\beta(t, \delta t)^{-2}\,z. \tag{82}$$

We map this score back to the manifold via the adjoint differential of the logarithm map,

$$s^{(\text{teach})}(\psi, \phi, t, \delta t) := (d\log_\phi)_\psi^*\, s_z^{(\text{teach})}(z, t, \delta t), \qquad z = \log_\phi(\psi), \tag{83}$$

which yields a tangent vector in $T_\psi\mathcal{M}$.

**Distillation into a global score model.**   We distill these analytic local conditional scores into a global score estimator $s_\theta(\psi, t)$ by minimizing a Riemannian denoising score matching objective,

$$\mathcal{L}(\theta) = \mathbb{E}\Big[\lambda(t, \delta t)\,\big\|\mathcal{P}_{\psi_t}\big(s_\theta(\psi_t, t)\big) - s^{(\text{teach})}(\psi_t, \psi_{t-\delta t}, t, \delta t)\big\|_{\text{FS}}^2\Big], \tag{84}$$

where $\mathcal{P}_\psi$ projects onto the tangent space and $\lambda(t, \delta t)$ is a variance-based weighting (we use $\lambda(t, \delta t) = \beta(t, \delta t)^2$ by default, matching standard VP denoising score matching).

## H.4. Choice of Prior Distribution

A practical diffusion model requires a tractable prior $p_T$ on $\mathbb{CP}^{d-1}$. Haar-random priors are theoretically appealing but expensive to sample and implement. SSDMs therefore adopt hardware-friendly priors such as Clifford $t$-design ensembles, which efficiently approximate Haar statistics while remaining implementable on near-term devices. From the perspective of our Riemannian OU construction, such priors provide convenient "endpoints" for reverse-time integration while preserving rotational symmetry properties that align with the FS geometry. Another promising direction is to use dissipative fixed-point ensembles induced by simple Lindblad dynamics as priors, which would further strengthen the physical interpretability of the forward process and may improve sampling efficiency.

## H.5. Classical vs. Quantum Score Models

SSDMs are agnostic to the parameterization of the score estimator $s_\theta(\psi, t) \approx \nabla_{\text{FS}} \log p_t(\psi)$. Classical neural networks provide strong baselines and enable rapid prototyping, while quantum neural networks (QNNs) may offer advantages when score features are naturally expressed via quantum measurements or when one seeks end-to-end quantum pipelines. Importantly, the primary contribution of SSDMs is the formulation of an intrinsic score-based generative framework on the quantum pure-state manifold; establishing a definitive advantage of QNN parameterizations remains an open question. A practically useful intermediate direction is hybrid modeling, where a classical network predicts a tangent-vector score from measurement features (e.g., overlaps or shadow-based estimates), while a QNN provides a structured inductive bias.

## H.6. QML Utility: Generative Augmentation in Quantum Representation Space

SSDMs enable representation-level augmentation: rather than generating new raw inputs, we generate additional quantum feature states that follow the learned pure-state ensemble distribution. Consider a downstream predictor $h$ (classical or quantum) trained on quantum representations. A standard supervised objective is the empirical risk

$$\min_{h \in \mathcal{H}}\; \frac{1}{n}\sum_{i=1}^n \ell\big(h(\psi(x_i)), y_i\big). \tag{85}$$

SSDM-generated states can be incorporated in two complementary ways.

**(i) Conditional generation (when available).**   If one trains class-conditional SSDMs $p_0(\psi \mid y)$, then additional samples $\tilde{\psi}_j \sim p_0(\psi \mid y)$ can be used for standard supervised augmentation.

**(ii) Unconditional augmentation via representation consistency.**   Even without labels for generated samples, SSDM augmentation can improve representation learning by enforcing prediction consistency under generated variations in the quantum feature space. Concretely, one may optimize

$$\min_{h \in \mathcal{H}} \frac{1}{n} \sum_{i=1}^{n} \ell\big(h(\psi(x_i)), y_i\big) + \alpha \, \mathbb{E}_{\tilde{\psi} \sim p_0}\Big[\mathrm{D}\big(h(\tilde{\psi}), \, h(\Pi(\tilde{\psi}))\big)\Big], \tag{86}$$

where $\mathrm{D}(\cdot, \cdot)$ is a divergence (e.g., squared error or KL) and $\Pi$ denotes a stochastic perturbation or pairing in the representation space. This viewpoint emphasizes SSDMs as a generative prior over $\mathbb{CP}^{d-1}$, enabling augmentation even when assigning reliable labels to generated states is difficult.

# I. Limitations and Future Directions

Our work has several limitations. First, our local-time teacher signal relies on the *local Euclidean OU/VP approximation* of the intrinsic forward diffusion in Fubini–Study normal coordinates. This approximation is most accurate in the small-$\delta t$ regime and in neighborhoods where normal-coordinate distortions are small. Although the leading-order OU teacher yields an analytic and stable training signal, residual curvature and volume-element effects enter at higher order (e.g., $O(\|z\|^2)$ terms in normal coordinates) and may introduce bias when $\delta t$ is not sufficiently small or when the diffusion explores regions with stronger curvature effects. Developing multi-scale teacher signals, curvature-aware corrections, or more accurate local approximations (e.g., higher-order normal-coordinate expansions or learned local transition models) could further enhance training stability and sample quality.

Second, our current framework focuses on *pure states* on $\mathbb{CP}^{d-1}$. Extending SSDMs to *mixed states* would require diffusion processes on the space of density matrices and appropriate notions of scores under quantum information geometry, which is nontrivial but important for modeling noisy quantum data and open-system dynamics.

Third, scalability remains a shared challenge for generative modeling of quantum pure states. While SSDMs can in principle be trained beyond $n = 6$ qubits using classical score networks, we observe that, as the number of qubits increases, generative quality metrics such as fidelity and MMD degrade rapidly. Importantly, this behavior is not specific to SSDMs: all baseline methods considered exhibit a similar or more severe performance drop as system size grows, with SSDMs remaining comparatively more robust in this regime. We believe this phenomenon reflects a fundamental difficulty of pure-state generative modeling in high-dimensional Hilbert spaces. As the number of qubits increases, the geometry of $\mathbb{CP}^{d-1}$ becomes increasingly complex, and the effective information captured by finite-capacity score models diminishes, making accurate learning of the ensemble distribution progressively harder. For this reason, and consistent with prior work in quantum generative modeling, existing studies—including ours—typically report results only for small numbers of qubits. Developing architectures and learning principles that better exploit structure in high-dimensional quantum state spaces remains an important direction for future research.

Finally, our framework implicitly assumes a degree of access to quantum state information during training. In particular, SSDMs operate from an ensemble-level perspective, where statistical information about quantum pure states is assumed to be available, either through repeated measurements, observable statistics, or effective descriptions induced by physical processes. While this assumption departs from a fully operational view of quantum state preparation and measurement, it reflects a deliberate trade-off: by learning a classical generative model of quantum state ensembles, one can replace repeated or expensive quantum state preparation with efficient classical sampling at the representation level. This perspective is especially relevant in near-term settings where quantum resources are limited, and classical generative surrogates may provide a practical alternative for augmenting or simulating quantum representations before large-scale fault-tolerant devices become available.

Despite these limitations, SSDMs establish a physically grounded and geometrically principled route for score-based generative modeling and sampling of quantum pure-state ensembles. We hope this framework will stimulate further research at the intersection of diffusion models, stochastic quantum dynamics, and QML.

