# OpenReview forum: "Stochastic Schrödinger Diffusion Models for Pure-State Ensemble Generation"
_ICML.cc/2026/Conference — Submitted to ICML 2026_

### Official Review · Reviewer_CZ5D · 2026-03-05

**Soundness:** 3
**Presentation:** 2
**Significance:** 2
**Originality:** 2
**Overall Recommendation:** 4
**Confidence:** 2

**Summary:**

This paper propose Stochastic Schrodinger Diffusion Models with Fubini-Study (FS) metric on complex-projective space. For the diffusion process, this paper used Riemannian diffusion and utilize Riemannian scores. Overall, this paper well-combined existing classical methods and manifold diffusion frameworks to the complex projective space.

**Compliance With Llm Reviewing Policy:**

Affirmed.

**Final Justification:**

This paper provides theoretical analysis on pure-state generation on quantum systems, while the experimental setups are not explained in detail, and limited to few qubit scale. However, as authors pointed out at the rebuttal, the contribution does not rely on the experiments but theoretical analysis, proposing the SSDM.

**Key Questions For Authors:**

On top of the weakness written above, I have several questions.

* What is the target distribution of the model? It does not explain which classical data is used to generate quantum feature map. Also, How did the authors trained "a shallow parameterized quantum circuits" to encode into n-qubit states? How many additional data is generated to train the SVM in RQ4? Experiment settings are not well provided.

* What is the Euclidean baseline for RQ2? Euclidean VP-SDE seems like classical algorithm. How will the diffusion model work with Euclidean SSDM on normalized states?

* Instead of QDDPM and QGAN, is there any baseline performance of quantum generative models?

**Limitations:**

yes, checked in the appendix.

**Strengths And Weaknesses:**

## Strength
- proposing method to generate samples from complex projection space using ssdms and score-based generative modeling framework
- utilizing Riemannian OU instead of Euclidean OU

## Weakness
- the settings of the experiments are hard to follow. How many samples are used to train the model? What is the circuit architecture (ansatz)?
- lack of implementation detail
- combination of existing works to apply well on quantum machine learning
- Why classical data on quantum circuit? QDDPM utilizes quantum data, and QGAN focuses on distribution loading for further advantage on QAE. Why do we need this model for classical data?
- Lack of baselines for quantum generative models such as QCBM.

---

> ### Author Rebuttal · Authors · 2026-03-29
>
> [W1&W2]
>
> We agree that the experimental setup was not sufficiently clear and will revise the manuscript accordingly.
>
> * Training data. We follow the same protocol as QDDPM, generating a cluster of quantum states around $|0\cdots0\rangle$ via small complex Gaussian perturbations with normalization:
>   $$
>   \psi = \frac{|0\cdots0\rangle + \epsilon \cdot \xi}{|||0\cdots0\rangle + \epsilon \cdot \xi||}.
>   $$
>   All data are simulated as normalized statevectors (no hardware or measurement). We sample mini-batches of size 64 from a pool of 4096 states.
>
> * Training configuration. The model is trained for 10,000 steps using AdamW (learning rate $2\times10^{-4})$. We use 500 diffusion steps and generate 256 samples for evaluation.
>
> * Score model. The score function is parameterized by a classical MLP: the input is the concatenation of real and imaginary parts of the state (dimension $2d$) together with a 128-dimensional time embedding. The network has 5 fully connected layers with hidden size 512 and SiLU activations. The output is projected onto the tangent space to respect the manifold geometry.
>
> * For comparison, the baselines use quantum circuits: QGAN uses a 120-layer PQC generator and a 16-layer PQC discriminator; QDDPM uses time-dependent PQCs with 6 layers per step over (T=20).
>
> [W3]
>
> We respectfully disagree that the work is merely a combination of existing ideas.  Prior Riemannian diffusion works do not address the quantum pure-state setting with projective geometry and phase invariance, while existing quantum diffusion methods typically focus on channel inversion, denoising, or circuit-based generation rather than intrinsic score-based modeling of ensemble distributions. Our key contribution is to bridge these directions through (i) an intrinsic forward diffusion on $\mathbb{CP}^{d-1}$ with an SSE realization, (ii) reverse-time dynamics driven by the Fubini--Study Riemannian score, and (iii) a local-time analytic teacher in FS normal coordinates that enables training without tractable transition densities. We believe this combination is nontrivial and specific to the quantum pure-state setting.
>
> [W4]
>
>
>
> We would like to clarify that our model generates quantum states, not classical data. In our experiments, although the initial states are constructed via classical simulation, they are treated as normalized quantum statevectors in $\mathbb{C}^{2^n}$. The model learns and generates distributions directly in the quantum state space (i.e., on $\mathbb{CP}^{d-1}$), rather than in the classical input space. Therefore, our setting is consistent with quantum generative modeling. The use of simulated data is solely for controlled evaluation and does not change the nature of the problem.
>
> [W5]
>
> QCBM models measurement outcome distributions using parameterized quantum circuits, i.e., classical probability distributions obtained from quantum measurements.
>
> In contrast, our work focuses on generative modeling of  pure states on $\mathbb{CP}^{d-1}$. Due to this difference in modeling targets and representation spaces, QCBM is not directly comparable. We will clarify this  in the revised version.
>
> [Q1]
>
> (i) The target distribution is a distribution over quantum states (pure states on $\mathbb{CP}^{d-1}$). No classical dataset  is used. All training data are quantum states generated via classical simulation (statevectors), and the model operates directly in quantum state space. ( See W1&W2)
>
> (ii) For RQ4, we use a two-class quantum-state dataset generated directly in Hilbert space. The training set contains 200 samples (100 per class), and the test set contains 400 samples (200 per class). We generate 200 additional states, pseudo-label them by prototype similarity, and retain the top 50\% most confident samples, yielding about 100 augmented samples for downstream training.
>
> [Q2]
>
> The Euclidean baseline in RQ2 applies a standard VP-SDE in $\mathbb{R}^{2d}$ by representing quantum states via their real and imaginary parts. The score model is trained with classical denoising score matching and does not enforce any quantum-state geometry.
>
> During sampling, vectors are first generated in $\mathbb{R}^{2d}$ and then mapped back to complex vectors and normalized to unit norm to obtain valid quantum states. Therefore, diffusion is not performed on normalized states, but in Euclidean space with post-hoc normalization.
>
> [Q3]
>
>
>  QDDPM and QGAN represent two main paradigms of quantum generative models, namely diffusion-based and adversarial approaches, which are most closely related to our setting. In addition, we include classical diffusion baselines, namely VP-SDE and RSGM. This further highlights the advantage of modeling diffusion intrinsically on the quantum state manifold.
>
> Other models such as QCBM focus on learning measurement probability distributions via parameterized quantum circuits, rather than modeling distributions over quantum states. We will clarify this distinction and expand the discussion  in the revised version.

---

> > ### Author Rebuttal · Reviewer_CZ5D · 2026-04-02
> >
> > Thank you for the detailed rebuttal.
> >
> > I now better understand the paper’s intended contribution and novelty, and I will update my score from 2 to 3. I also recognize that part of my initial reading was inaccurate, so I will correspondingly reduce my confidence score.
> >
> > That said, I still have question regarding the experimental setup.
> >
> > My current understanding is that the experiments are conducted in a fully quantum setting and that generating pure states is the primary objective. My earlier confusion about classical data came from Appendix G, which states that (x) is sampled from classical data and then embedded into quantum feature states via shallow parameterized quantum circuits. This seems somewhat at odds with the clarification given in the rebuttal. Also, if the experimental setting is entirely based on quantum states, how was the Euclidean VP-SDE baseline trained and evaluated?
> >
> > I would appreciate further clarification on this point, as it would help me better assess the consistency of the experimental setup.

---

> > > ### Author Response · Authors · 2026-04-03
> > >
> > > Thank you for the follow-up. We agree that the source of confusion is our presentation, and we appreciate the opportunity to clarify this point more precisely.
> > >
> > > The key distinction is between (i) how the target quantum-state ensemble is constructed, and (ii) what object the generative model is trained to model.
> > >
> > > In all experiments, the generative model is trained to model a distribution over normalized pure quantum states. However, the target pure-state ensemble is obtained in two different ways in different parts of the paper.
> > >
> > > For the main generative experiments (RQ1–RQ3), the dataset is a synthetic pure-state ensemble constructed directly in Hilbert space. Concretely, we generate normalized statevectors by perturbing the reference state $|0\cdots 0\rangle$ with small complex Gaussian noise and then renormalizing. Thus, these experiments do not start from a classical dataset $x$; they directly use quantum states as training samples.
> > >
> > > For the QML augmentation setting (RQ4 / Appendix G), the setup is different: we first construct a labeled raw-input dataset $x$, and then encode those inputs into quantum feature states $|\psi(x)\rangle$. The generative model is then applied in the quantum representation space, i.e., it models and generates quantum states rather than the original raw inputs. Therefore, raw/classical data appears only as a way of constructing one particular target pure-state ensemble; the learned generative model still operates on quantum states.
> > >
> > > Regarding the Euclidean VP-SDE baseline in RQ2, it is trained on the same target quantum-state samples as SSDM; the difference is only the representation used during diffusion modeling. Each normalized quantum state $\psi$ in $\mathbb{C}^d$ is represented extrinsically as a real vector in $\mathbb{R}^{2d}$ by concatenating its real and imaginary parts. A standard Euclidean VP-SDE with denoising score matching is then trained on these $\mathbb{R}^{2d}$ vectors. Thus, the Euclidean baseline is not trained in the original classical input space; it is trained on the same quantum states, but in an ambient Euclidean coordinate representation rather than intrinsically on $\mathbb{CP}^{d-1}$.
> > >
> > > During sampling, the Euclidean baseline first generates vectors in $\mathbb{R}^{2d}$. These vectors are then mapped back to complex vectors in $\mathbb{C}^d$ and normalized to unit norm, yielding valid pure quantum states. These generated normalized states are then evaluated against the target pure-state ensemble using the same evaluation protocol as SSDM, i.e., the same quantum-state metrics reported in the paper.
> > >
> > > So the apparent inconsistency comes from the fact that Appendix G used a broader description that covered both settings, whereas our previous rebuttal described only the main synthetic pure-state setting. We agree that this distinction was not made explicit enough, and in the revision we will separate these two settings explicitly and state, for each experiment, (i) how the state dataset is constructed, (ii) what object is modeled, and (iii) in which space each baseline is trained.

---

### Official Review · Reviewer_uKVx · 2026-03-10

**Soundness:** 3
**Presentation:** 3
**Significance:** 2
**Originality:** 3
**Overall Recommendation:** 4
**Confidence:** 3

**Summary:**

This paper proposed a specific method to do generative modelling of complex vector representations of pure quantum states based on an original version of Riemannian diffusion model. The paper focuses on the description of the model design, including forward and backward diffusion equations. The highlight of the model architecture is that it uses local Euclidean approximation to build a local approximated teacher score to overcome the intractability issue of Riemannian score. The paper also conducts small experiments with 2,4,6 qubits to show that the proposed method help improve the generative quality and downstream task of quantum kernel over baseline models of their choice.

**Compliance With Llm Reviewing Policy:**

Affirmed.

**Final Justification:**

Strength: clear presentation, moderate novelty, interesting and solid theory.
Weakness: missing details on experiment setup, clear limitation in method scalability and application value.

The rebuttal addressed my main concerns, by supplementing details on experiment setup and adding one small ablation study. Also, the authors made fair assessment on the cause of limited scalability and challenge. I encourage the authors to make this part clear in the revised paper as a slightly negative message to inspire future studies in the field of quantum diffusion models to focus on scalability. After rebuttal, I change my suggestion from weak reject to weak accept.

**Key Questions For Authors:**

1. Could you please include in the manuscript a complete description of the proposed score network, or at least the ones used in the experiments, and all baseline models, including architecture, input/output form, embedding method, hyperparameters for training and inference, ...?

2. Could you please clarify what is the scalability bottleneck in implementation and experiments? What prevents you from going above 6 qubits? Is the limitation coming from data representation, representation of score fields, something specific to Riemannian diffusion, or something else?

3. Could you please compare your approach against the stand data augmentation workflow in QML, i.e., augmenting in raw data space, and then push through quantum feature map? (which clearly scales beyond 6 qubits, even to beyond-classical numbers of qubits on real quantum hardware already) A fair comparison on downstream task performance and resource cost would be appreciated.

4. You mentioned that quantum neural network can also potentially be used as the score network. Can you please analyse what is the advantage / disadvantage of doing this instead of classical NN in terms of training efficiency, sampling efficiency, scalability, ...?

**Limitations:**

No. No obvious negative social impact is noticed from this work.

**Strengths And Weaknesses:**

This paper is very clearly written and the messages conveyed is clear. It contains detailed derivation of the chosen forward and backward equations and explains the reasoning behind the choices. On the theoretical side, the analytical details provide are sufficient to support the correctness and feasibility of the model.

In contrast, the report on implementation and experiments is currently relatively weak. The exact architecture and implementation details of the proposed models and the baseline models compared are not reported in the manuscript, though part of the implementation and experiment pipeline are provided in the supplementary files. A description of the model architecture and training/sampling hyperparameters should at least be shown in the manuscript for clarity.

The scalability of the proposed model is of big concern. Current experiments stop at 6 qubits, reasons remaining unknown. In the best case of using the full Hilbert space, that only translates into 2^6 = 64 dimensional classical data, which is really small considering that diffusion models today can process huge classical data like video clips or 3d engineering data. The authors need to explain what exactly is the limitation on scalability. The best way of doing this would be to describe in detail in the manuscript the architecture, what encoding is used, what architecture is used, and how training (optimization) is done exactly and make an analysis what is the computational bottleneck. Is it the encoding of complex vectors, or fundamental limitation of Riemannian diffusion?

The current experiments report better performance over other quantum diffusion models, which is plausible. However, as the manuscript repeatedly mentions the application to QML, an equally interesting question is, how does the downstream task performance compare to the default way of data augmentation, i.e., first generating new raw data, and then do quantum embedding via feature map. As a side note, the default approach can clearly scale beyond 6 qubits and allow for QML experiments with 10-20 qubits on one or a few GPUs.

In conclusion, the paper is interesting from a mathematical perspective. However, to make it more impactful on the community, it is better to have at least one of the following things:
1. Detailed analysis on the computational efficiency of each component of the architecture and complete reasoning on the limitations / opportunities for scalability for realistic applications. (This can largely improve the paper even if the conclusions are negative for the proposed method.)
2. Demonstration of comparable / superior performance over the default way of data augmentation in QML. Please ensure that both cases use the same quantum feature map and also compare the resource cost.

---

> ### Author Rebuttal · Authors · 2026-03-31
>
> [W1 & Q1 experimental setup]
>
>  We agree that the experimental setup and implementation details were not sufficiently clear in the main manuscript, and we will revise accordingly to improve clarity and reproducibility.
>
> Training data and score network. Details of the training data generation and score network architecture are provided in our response to Reviewer CZ5D’s [W1 & W2], and will be incorporated into the main manuscript.
>
> Training and diffusion hyperparameters. The model is trained for 10,000 steps using AdamW (learning rate (2\times10^{-4}), gradient clipping at 1.0). We use a diffusion horizon (T=1) with 500 discretization steps and a local-time step (\delta t = 1/500). The noise schedule follows an exponential interpolation between (\sigma_{\min}=0.05) and (\sigma_{\max}=1.0), and the OU drift strength is set to (\lambda=0.2). The loss uses variance-based weighting (\lambda(t,\delta t)=\beta(t,\delta t)^2), consistent with VP-style score matching.
>
> Sampling configuration. Reverse-time sampling is performed with 500 steps using an Euler–Maruyama discretization on the manifold, and 256 samples are generated for evaluation.
>
> Baseline models. For fair comparison, we follow standard implementations of prior work. QGAN uses a 120-layer PQC generator and a 16-layer PQC discriminator; QDDPM uses time-dependent PQCs with 6 layers per step (with (T=20)). All baselines are evaluated under the same statevector simulation setting.
>
> We will include a complete and self-contained description of these details in the revised manuscript.
>
>
> [W2&Q2 the scalability bottleneck]
>
>
> While a 6-qubit system corresponds to a 64-dimensional complex vector space, quantum states lie on the constrained manifold $ \mathrm{CP}^{d-1} $ due to normalization and global phase, rather than in Euclidean space. This induces a nonlinear geometry that must be strictly preserved during modeling and sampling.
>
> Unlike classical data (e.g., images), where structure is captured via soft inductive biases, quantum states are subject to hard constraints, making generative modeling inherently more challenging.
>
> The limitation to 6 qubits is therefore not due to GPU memory, but reflects this intrinsic difficulty. As the number of qubits increases, generation quality (e.g., fidelity, MMD) degrades significantly. This degradation is not specific to SSDMs—baseline methods such as QGAN and QDDPM exhibit even more severe drops. Likewise, Euclidean VP-SDE models perform poorly in quantum-state-like settings, achieving low fidelity even for small systems (e.g., 2 qubits), suggesting the challenge is not method-specific.
>
> In addition to exponential scaling, the lack of strong inductive biases for quantum states further limits performance.
>
> Overall, the scalability issue stems from the intrinsic difficulty of generative modeling on high-dimensional quantum state manifolds $ \mathrm{CP}^{d-1} $, as also evidenced by prior work (e.g., QDDPM) being restricted to very small systems.
>
> [W3&Q3 the stand data augmentation]
>  Comparison with standard input-space augmentation
>
> | Qubits | Method                         |      Acc ↑ | Kernel Alignment ↑ | Kernel Gap ↑ | Mean Margin ↑ |
> | ------ | ------------------------------ | ---------: | -----------------: | -----------: | ------------: |
> | 6      | Original only                  |     0.6475 |             0.0831 |       0.0039 |    **0.0559** |
> | 6      | Raw VP diffusion (input-space) |     0.6375 |             0.0622 |       0.0014 |        0.0540 |
> | 6      | **SSDM augmentation (ours)**   | **0.6550** |         **0.1053** |   **0.0081** |        0.0518 |
> | 8      | Original only                  |     0.5875 |             **0.0726** |       0.0004 |    **0.0700** |
> | 8      | Raw VP diffusion (input-space) |     0.5950 |             0.0622 |       0.0014 |        0.0540 |
> | 8      | **SSDM augmentation (ours)**   | **0.5975** |             0.0696 |   **0.0016** |        0.0579 |
>
> We have added a comparison with the standard QML augmentation pipeline, i.e., augmenting samples in the raw input space followed by the same quantum feature map. As shown in the table, our representation-level augmentation is consistently competitive or better at both 6 and 8 qubits. These results indicate that improvements in the classical input space do not necessarily translate into better quantum representations after encoding.
>
> [Q4 QNNs]
>
> QNNs could in principle enable quantum advantage by leveraging native quantum operations. However, under current NISQ constraints, they suffer from noise, limited circuit depth, and significant measurement overhead, leading to inefficient training and costly score evaluations during sampling. Therefore, we adopt classical neural networks as a more practical choice and leave QNN-based implementations for future work.

---

> > ### Author Rebuttal · Reviewer_uKVx · 2026-04-01
> >
> > The authors have addressed most of my questions. Although the scalability and application value of the proposed method are still of big concern, the paper provides interesting and valuable theoretical contribution. In the rebuttal, the authors are also transparent about the scalability challenge, and I hope the authors can clearly present the limitation and challenge (of the proposed method, and the subject of quantum diffusion models in general) in the revised paper. I decide to increase my score  from 3 to 4.

---

> > > ### Author Response · Authors · 2026-04-03
> > >
> > > Thank you for the constructive feedback and for increasing your score. We agree that scalability and practical applicability are important challenges, and we will make these limitations clearer in the revised paper, both for our method and for quantum diffusion models more broadly.

---

### Official Review · Reviewer_MxTh · 2026-03-13

**Soundness:** 3
**Presentation:** 3
**Significance:** 2
**Originality:** 3
**Overall Recommendation:** 4
**Confidence:** 2

**Summary:**

This paper proposes Stochastic Schrödinger Diffusion Models (SSDMs), a framework
  for generative modeling of quantum pure states. The key idea is to do diffusion
  modeling directly on the pure-state manifold CP^{d-1} with the Fubini-Study metric,
  rather than treating quantum states as Euclidean vectors. Since score computation
  on this manifold is intractable, the main technical contribution is a local-time
  training trick that exploits the fact that short-time transitions look Euclidean
  in normal coordinates. Experiments on 2, 4,
  and 6 qubit systems show SSDMs outperform baselines on distribution-matching
  metrics, and that respecting the geometry matters increasingly with system size.

**Compliance With Llm Reviewing Policy:**

Affirmed.

**Final Justification:**

Authors addressed my questions; I maintain my current score

**Key Questions For Authors:**

1. **Classical augmentation baseline:** Have you considered comparing against
     the straightforward alternative of augmenting at the classical input level
     (e.g., standard generative modeling or noise augmentation on x) and then
     encoding through the feature map? The claim that classical perturbations
     produce pathological quantum states is key to the motivation but is
     not empirically validated. If classical augmentation performs comparably
     on the downstream QML tasks in Section 5.4, it would significantly
     weaken the case for doing generative modeling in state space, no?

  2. **Scaling trajectory:** If the method
     cannot produce high-quality samples beyond a handful of qubits, what
     is the practical setting where SSDMs would be preferred over classical
     alternatives? The paper frames this as a shared limitation, but it
     seems to undermine the premise that generative modeling of quantum
     states is a tractable and useful problem.

  3. **Variance and reproducibility:** Table 1 does not report error bars or
     variance across runs. How stable are these results across random seeds
     and training runs? This is especially important at n=6 where absolute
     performance is modest.

**Limitations:**

yes

**Strengths And Weaknesses:**

**Strengths**

- **Technically non-trivial:** The core solution -- doing
    score-based diffusion on CP^{d-1} where transition densities are intractable --
    appears non-trivial. The paper tackles it with a principled geometric approach
    rather than ad hoc workarounds.

- **Convincing empirical results where tested:** SSDMs convincingly
    outperform baselines across metrics on the shown experiments.

**Weaknesses**

 - **Weak motivation:** The paper doesn't make a strong case for
    why generative modeling of pure-state ensembles is a problem we need to solve.
    The main downstream application is data augmentation for QML, but the
    evaluation there is thin. More importantly, the argument for doing generative
    modeling in quantum state space rather than at the classical input level is
    made rhetorically but never tested -- they claim classical perturbations
    produce pathological states, but don't compare against the obvious baseline
    of augmenting classically and then encoding.

  - **Scale undermines the premise:** Results only go up to 6
    qubits, and performance degrades substantially even over that range. The authors acknowledge this is a shared
    limitation across methods, but it cuts against the paper's framing: if the
    approach can't scale beyond small systems, and all the test ensembles are
    generated from classical data anyway, it's unclear when you'd use this
    in practice over classical alternatives.

---

> ### Author Rebuttal · Authors · 2026-03-31
>
> We thank the reviewer for the constructive feedback and address the key concerns below.
>
> [W1&Q1 Classical augmentation baseline]
>
> We agree that comparing with classical augmentation is important. Our motivation is that perturbations in input space can map to highly non-smooth or even orthogonal changes in quantum feature space, which may harm representation quality. While this effect is well-known in QML literature, we acknowledge that it is not explicitly validated in our current experiments.
>
>  Comparison with standard input-space augmentation
>
> | Qubits | Method                         |      Acc ↑ | Kernel Alignment ↑ | Kernel Gap ↑ | Mean Margin ↑ |
> | ------ | ------------------------------ | ---------: | -----------------: | -----------: | ------------: |
> | 6      | Original only                  |     0.6475 |             0.0831 |       0.0039 |    **0.0559** |
> | 6      | Raw VP diffusion (input-space) |     0.6375 |             0.0622 |       0.0014 |        0.0540 |
> | 6      | **SSDM augmentation (ours)**   | **0.6550** |         **0.1053** |   **0.0081** |        0.0518 |
> | 8      | Original only                  |     0.5875 |             **0.0726** |       0.0004 |    **0.0700** |
> | 8      | Raw VP diffusion (input-space) |     0.5950 |             0.0622 |       0.0014 |        0.0540 |
> | 8      | **SSDM augmentation (ours)**   | **0.5975** |             0.0696 |   **0.0016** |        0.0579 |
>
>
> We have added a comparison with the standard QML augmentation pipeline, i.e., augmenting samples in the raw input space followed by the same quantum feature map. As shown in the table, representation-level augmentation via SSDM is consistently competitive or better at both 6 and 8 qubits, especially in terms of kernel alignment and kernel gap, which more directly reflect representation quality. This provides empirical support for our motivation: augmenting directly in quantum state space can better preserve and enrich the structure of quantum representations, whereas input-space perturbations may not translate effectively after encoding.
>
> [W2&Q2 Scaling trajectory]
>
> We agree that scalability is a key limitation. This is not due to GPU constraints or the diffusion framework itself, but stems from the exponential complexity of representing and learning distributions over pure states in high-dimensional Hilbert spaces. As system size increases, both score estimation and sampling become significantly harder, leading to the observed degradation in generation quality. This limitation is shared by existing quantum generative models, indicating a fundamental challenge rather than a method-specific issue.
>
> At the same time, we emphasize that this research direction is still at an early stage. Our goal is not to claim immediate superiority over classical alternatives, but to explore generative modeling directly in quantum state space as a forward-looking direction. As quantum hardware and algorithms advance, we expect such approaches to become increasingly relevant for larger-scale quantum data that cannot be efficiently handled classically. We will clarify this positioning and provide a more detailed discussion of computational bottlenecks in the revision.
>
> [Q3 Variance and reproducibility]
>
> Thank you for pointing this out. Our results are averaged over multiple random seeds, but we agree that this is not clearly reported. We will include variance statistics (e.g., error bars across runs) and clarify experimental details to improve reproducibility.

---

> > ### Author Rebuttal · Reviewer_MxTh · 2026-04-05
> >
> > Authors addressed my questions; I maintain my current score

---

> > > ### Author Response · Authors · 2026-04-06
> > >
> > > Thank you for your feedback and thoughtful evaluation

---

### Official Review · Reviewer_cAsV · 2026-03-16

**Soundness:** 3
**Presentation:** 3
**Significance:** 3
**Originality:** 3
**Overall Recommendation:** 5
**Confidence:** 3

**Summary:**

The paper introduces Stochastic Schrödinger Diffusion Models (SSDMs), a novel intrinsic score-based generative framework designed to sample from quantum pure-state ensembles. Because quantum pure states reside on the non-Euclidean complex projective manifold $\mathbb{CP}^{d-1}$ endowed with the Fubini-Study (FS) metric, standard Euclidean diffusion models cannot be directly applied. To solve this, the authors construct a forward Riemannian diffusion process with a stochastic Schrödinger equation (SSE) realization. To circumvent the intractability of transition densities on this manifold, they introduce a local-time training objective utilizing a local Euclidean Ornstein-Uhlenbeck (OU) approximation in FS normal coordinates, yielding an analytic Gaussian teacher score. Empirical evaluations on up to 6 qubits demonstrate that SSDMs can successfully generate pure states that match target ensemble statistics (fidelity, MMD, observable mismatch, and entanglement metrics) and that these generated representations improve downstream quantum machine learning (QML) tasks via data augmentation.

**Compliance With Llm Reviewing Policy:**

Affirmed.

**Final Justification:**

I stay by my original score.

**Key Questions For Authors:**

1.
**Impact of numerical errors from tangent projections:** In practice, how sensitive is the empirical generative performance (e.g., fidelity and MMD) to the choice of $\delta t$, and do you observe significant numerical drift when integrating over the full diffusion horizon?


2.
**Scalability to higher qubit counts:** You honestly note the performance degradation beyond $n=6$ qubits. While this is a shared challenge across quantum generative models, could you elaborate on potential architectural or theoretical modifications that might help mitigate this bottleneck in future work?


3.
**Extension to mixed states:** You mention that extending SSDMs to mixed states/density matrices would require new diffusion processes and scores under quantum information geometry. Do you anticipate that the core machinery (specifically the local-time teacher score derived from an OU approximation) could be directly adapted to a manifold of density matrices, or would the geometric properties of mixed states fundamentally break this approximation?

**Limitations:**

Yes

**Strengths And Weaknesses:**

=== Strengths ===

- **Soundness:** The submission is technically rigorous and well-supported by theoretical derivations in the appendices. The formulation of the forward diffusion as a Riemannian process and its connection to physical stochastic Schrödinger equations is principled. Furthermore, the local Euclidean OU approximation in normal coordinates provides a mathematically sound workaround for intractable transition densities.

- **Presentation:** The paper is generally well-structured, clearly written, and easy to follow. The transition from physical motivation to geometric diffusion modeling is articulated well.

- **Originality:** While Riemannian score-based models and quantum generative models exist independently, combining them via an intrinsic formulation on the $\mathbb{CP}^{d-1}$ manifold using the Fubini-Study metric is a novel and creative approach.


=== Weaknesses ===

- As noted by the authors, the model struggles to scale beyond $n=6$ qubits due to the increasing complexity of the high-dimensional Hilbert space. Additionally, in practice, the method requires mapping ambient increments to the tangent space and retracting back to the manifold; while theoretically sound at infinitesimal steps, this projection might incur compounding numerical errors during practical implementation.

- There are a few minor presentation issues. The equations in Figure 1 are too small to read comfortably. Additionally, there is a minor typo in Section 5.2 ("Table 1 and summarizes the effect..."). A brief intuitive explanation of "universal geometric structure" early in the introduction could also benefit readers less familiar with Riemannian geometry.

---

> ### Author Rebuttal · Authors · 2026-03-31
>
> [W1] Clarifying numerical stability and implementation details
>
> We agree that scalability is limited by the increasing complexity of the high-dimensional Hilbert space, as discussed in the paper. Regarding the tangent projection and retraction steps, in practice we use small step sizes and perform updates in local orthonormal frames, which helps control numerical drift. Empirically, we do not observe significant instability within reasonable discretization regimes, although larger step sizes can degrade performance. We will clarify these implementation details and discuss numerical stability more explicitly in the revision.
>
> ---
>
> [W2] Improving presentation and accessibility
>
> Thank you for pointing these out. We will enlarge the equations in Figure 1 for better readability, fix the typo in Section 5.2, and add a brief intuitive explanation of the “universal geometric structure” in the introduction to improve accessibility.
>
> ---
>
> [Q1] Step size sensitivity and empirical stability
>
> In practice, we use small step sizes and perform updates in local orthonormal frames followed by retraction, which helps mitigate numerical drift. Empirically, we observe that performance (e.g., fidelity and MMD) is relatively stable within a reasonable range of δt, while excessively large steps degrade sample quality. We will include additional discussion and sensitivity analysis in the revision.
>
> ---
>
> [Q2] Scalability challenges and future directions
>
> As noted, scalability is primarily limited by the exponential complexity of modeling distributions over high-dimensional Hilbert spaces, rather than the diffusion framework itself. Potential directions to improve scalability include incorporating stronger structure (e.g., low-rank or circuit-based parameterizations), hybrid classical–quantum score models, and more efficient representations of quantum states. We will expand this discussion in the revision.
>
> ---
>
> [Q3] Extending to mixed states
>
> We believe the core idea—local-time score estimation via normal-coordinate approximations—can be extended in principle. However, the geometry of density matrices (e.g., Bures or quantum Fisher metrics) is more complex, and the OU approximation may require nontrivial modifications. We therefore view this as an important but nontrivial extension, and will clarify this point and outline possible directions in the revision.

---

> > ### Author Rebuttal · Reviewer_cAsV · 2026-04-04
> >
> > I thank the author for clarification. The response has resolved my questions. I will keep my original score.

---

> > > ### Author Response · Authors · 2026-04-06
> > >
> > > Thank you for your feedback and thoughtful evaluation.

---

### Decision · Program_Chairs · 2026-04-30

**Decision:**

Reject

**Comment:**

This paper proposes Stochastic Schrödinger Diffusion Models for the generative modeling of pure quantum states. In particular, the authors construct a forward Riemannian diffusion process on the complex projective manifold equipped with the Fubini-Study metric, and introduce a local approximation technique to implement it.

The reviewers agreed on the originality of the proposed method. Nevertheless, all reviewers expressed concerns about the scalability of the method beyond 6 qubits, which, according to the authors’ response, seems difficult to improve at present; as a result, the reviewers were conservative in recommending this paper. Moreover, all reviewers were concerned about the presentation of the paper; in particular, the experiments section needs revision.  Overall, the decision is borderline but may lean very slightly downwards given the general reluctance and no reviewer that wanted to "champion" the paper for acceptance.